# ReQAT: Achieving Full-Precision Reasoning Accuracy with 4-bit Floating-Point Quantization-Aware Training

**Janghwan Lee** [1]  **Sihwa Lee** [1]  **Jinseok Kim** [2]  **Yongjik Kim** [2]  **Jieun Lim** [2]  **Jinwook Oh** [2]  **Jungwook Choi** [1]

## Abstract

Large Reasoning Models (LRMs) achieve strong problem-solving through long chain-of-thought, but their deployment is constrained by the high cost of full-precision inference and growing KV cache footprints. Microscaled FP4 formats enable efficient FP4 deployment; however, fully quantizing weights, activations, and KV caches (W4A4KV4) causes severe reasoning degradation that existing PTQ and QAT fail to recover. We identify that FP4 failures concentrate on low-entropy tokens—precise symbolic commitments such as digits and operators—where quantization noise inflates sampling errors that cascade through reasoning traces. Based on this insight, we propose ReQAT, a reasoning-centric FP4 training framework with three components: (i) Trace-Aligned QAT (TAQ), which revisits identical reasoning traces to focus updates on critical low-entropy decisions; (ii) Selective Entropy Minimization (SEM), which reinforces confidence at low-entropy positions; and (iii) Q-FIT, a quantization-friendly initialization that jointly calibrates RoPE-consistent KV cache transformations to stabilize QAT. Under the same training budget, ReQAT not only recovers but surpasses BF16 fine-tuning accuracy, while delivering up to $3.9\times$ throughput speedup on NVIDIA DGX Spark and $3.1\times$ on B200.[1]

## 1. Introduction

Large Reasoning Models (LRMs) have emerged as a transformative paradigm for solving complex multi-step mathematical and logical problems by generating extensive chain-

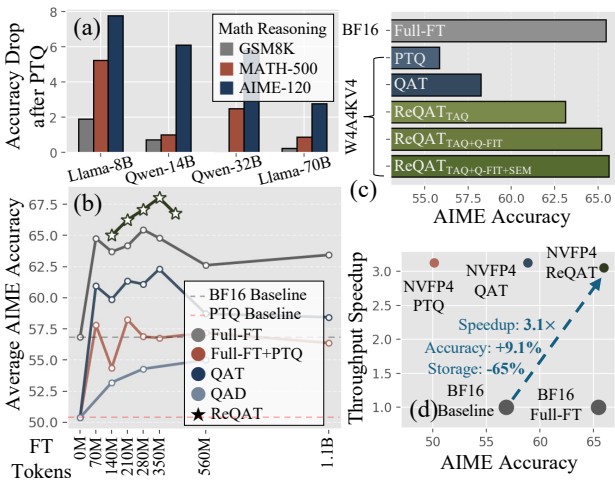

*Figure 1.* (a) NVFP4 W4A4KV4 PTQ accuracy drop across DeepSeek-R1-Distilled models. Accuracy of (b) MXFP4 W4A16 and (c) NVFP4 W4A4KV4 on R1-Qwen-14B, comparing Full-FT and QAT. (d) Accuracy–throughput across deployment recipes.

of-thought (CoT) traces (OpenAI et al., 2024; Guo et al., 2025; Yang et al., 2025; Guha et al., 2025). These models rely heavily on *inference-time scaling*, allocating additional compute during sequential decoding to solve increasingly difficult instances. However, deploying such scaling at the throughput required for modern serving—processing thousands of concurrent requests—presents a formidable efficiency gap. This cost grows jointly along three critical axes: (i) high memory-bandwidth demands for repeated weight loading during autoregressive decoding (Lin et al., 2025; 2024), (ii) massive aggregate floating-point operations (FLOPs) (Zhao et al., 2024), and (iii) a key-value (KV) cache footprint that scales linearly with sequence length, often exceeding 16K tokens in long-horizon reasoning tasks (Song et al., 2025; Zhang et al., 2025d). Together, memory traffic, compute, and KV storage create a critical efficiency gap that limits practical deployment of LRMs.

To bridge this gap, the industry is transitioning toward microscaled 4-bit floating-point (FP4) arithmetic, which pairs aggressive compression with hardware-native acceleration. Formats such as MXFP4 and NVIDIA's NVFP4 employ an E2M1 layout (2-bit exponent, 1-bit mantissa) to main-

---

[1]Hanyang University, [2]Rebellions Inc., Republic of Korea. Correspondence to: Jungwook Choi <choij@hanyang.ac.kr>.

*Proceedings of the $43^{rd}$ International Conference on Machine Learning*, Seoul, South Korea. PMLR 306, 2026. Copyright 2026 by the author(s).

[1]The project repository is available at https://github.com/aiha-lab/ReQAT.

tain numerical expressiveness through fine-grained scaling (Rouhani et al., 2023; Alvarez et al., 2025; AMD, 2025). This transition is catalyzed by native hardware support such as that in the NVIDIA Blackwell architecture, where B200 Tensor Cores deliver up to 9 PFLOPS under FP4 inference—approximately $4\times$ higher than FP16—and provide dedicated support for fully quantized NVFP4 W4A4KV4 (i.e., using FP4 for all weights, activations, and KV caches) inference (Nvidia, 2024). These advancements are not limited to hyperscale datacenters but extend to compact edge systems like NVIDIA DGX Spark (Nvidia, 2025), making FP4 the standard for cost-effective reasoning inference across diverse deployment scales.

Despite the hardware advances, aggressive quantization to W4A4KV4 configurations typically results in substantial accuracy degradation. Standard post-training quantization (PTQ) pipelines (NVIDIA, 2026) yield large accuracy drops across distilled reasoning models, as illustrated in Fig. 1(a). While quantization-aware training (QAT) is often viewed as a reliable mechanism for recovering PTQ-induced accuracy loss, existing recipes frequently fail to close the gap to the bfloat16 (BF16) baseline, even when scaling the fine-tuning token budget. As illustrated in Fig. 1(b), both QAT (Liu et al., 2025b) and quantization-aware distillation (QAD) (NVIDIA, 2026) improve over PTQ baselines, yet remain notably below BF16 full fine-tuning (FT) accuracy. The challenge intensifies once the KV cache is quantized, where channel-wise outliers and the rotational structure of Rotary Positional Embeddings (RoPE) introduce layer-dependent distortions. Existing fixed smoothing or shifting strategies fail to adapt to these oscillating token statistics (Lin et al., 2025; Baek et al., 2025; Zhang et al., 2025a), leaving a persistent performance gap (Fig. 1(c)).

In this work, we propose ReQAT, a reasoning-centric training framework motivated by the novel insight that LRM quantization failures concentrate on low-entropy tokens. Our analysis reveals that precise symbolic commitments—such as digits and operators—are highly sensitive to quantization noise, whereas high-entropy connective phrases remain robust. ReQAT addresses this through three key innovations: (i) Trace-Aligned QAT (TAQ), a two-stage procedure that revisits identical reasoning traces to focus updates on critical low-entropy decisions rather than trace variability; (ii) Selective Entropy Minimization (SEM), a new auxiliary loss that reinforces model confidence specifically at low-entropy positions; and (iii) Q-FIT, a quantization-friendly initialization that jointly calibrates pre-RoPE scaling and post-RoPE shifting to stabilize the KV cache across layer-specific distributions.

We evaluate ReQAT across multiple LRMs and challenging benchmarks including AIME-120, MATH-500, and GSM8K. Our results demonstrate a significant milestone:

ReQAT frequently surpasses BF16 full FT accuracy while utilizing the same training budget, for instance, achieving $65.94\%$ AIME accuracy under NVFP4 W4A4KV4 compared to $56.83\%$ for BF16 baseline and $65.46\%$ after full FT (Fig. 1(c)). Furthermore, our implementation on real-world NVIDIA Blackwell systems using TensorRT-LLM demonstrates substantial end-to-end serving throughput improvements, achieving a $3.1\times$ throughput speedup on B200 and $3.9\times$ on DGX Spark relative to BF16 baselines. These results establish ReQAT as a practical pathway to efficient, high-throughput LRM inference on production hardware.

## 2. Background

### 2.1. Quantization for Efficient Inference

**Microscaled FP4 formats.** MXFP4 and NVFP4 are widely used microscaled FP4 formats for efficient LLM inference, both adopting an E2M1 layout with different scaling granularities (detailed comparison is in Table 8 in Appendix). Common deployment settings include MXFP4 W4A16 (OpenAI et al., 2025), MXFP4 W4A4 (AMD, 2025; Rouhani et al., 2023), and NVFP4 W4A4KV4 (Nvidia, 2024), which additionally quantizes the KV cache. These configurations form the primary FP4 inference regimes considered in this work.

**Post-training quantization.** Several PTQ methods developed for LLMs have been applied to LRMs, including AWQ (Lin et al., 2024) for W4A16 and QuaRot (Ashkboos et al., 2024) and FlatQuant (Sun et al., 2025) for W4A4KV4 settings. However, prior work reports substantial reasoning accuracy degradation under W4A4KV4 quantization for LRMs (Liu et al., 2025a), motivating the need for QAT.

**KV cache quantization via transformation.** KV cache quantization has been studied for long-context LLMs (Liu et al., 2024; Hooper et al., 2024), but remains challenging due to channel-wise outliers in the key cache (Lin et al., 2025). Several methods reduce KV cache quantization error by applying transformations that modify the key distribution without changing the functionality of RoPE or attention computation (Lin et al., 2025; Zhang et al., 2025b). Let $Q^{\mathrm{pre}}, K^{\mathrm{pre}} \in \mathbb{R}^d$ denote pre-RoPE queries and keys, where RoPE applies a rotation to channel pairs $(r, r + d/2)$. Attention is computed as $z = \mathrm{softmax}\left(\frac{1}{\sqrt{d}} Q\hat{K}^{\top}\right)\hat{V}$, where $Q = \mathcal{R}(Q^{\mathrm{pre}})$ and $\hat{K}, \hat{V}$ denote quantized KV caches. One approach applies paired channel-wise scaling before RoPE (Lin et al., 2025), using a vector $s$ satisfying $s_r = s_{r+d/2}$, resulting in scaled queries and keys that preserve the attention computation. Another approach applies channel-wise shifting after RoPE (Zhang et al., 2025b), $\tilde{K} = \mathcal{R}(K^{\mathrm{pre}}) - m$, leveraging the shift-invariance of the softmax. These transformations use fixed parameters during inference.

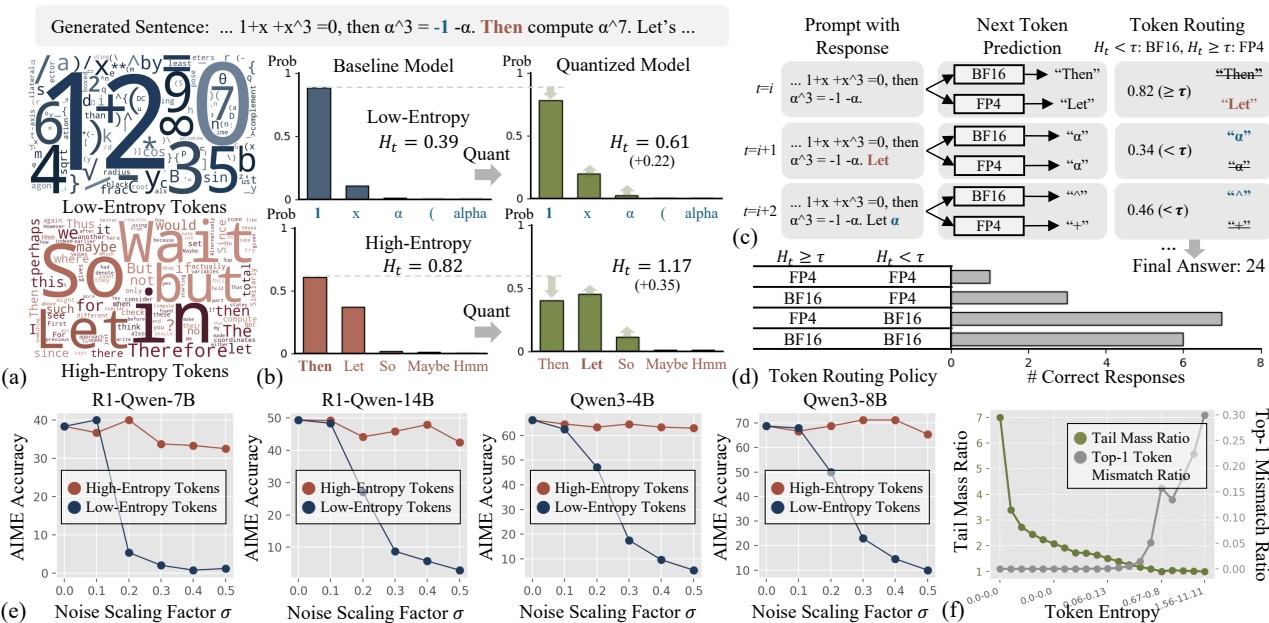

*Figure 2.* (a) Visualization of low- and high-entropy tokens. (b) Changes in token probability distributions under FP4 quantization. (c-d) Illustration of entropy-aware mixed-precision decoding. (e) Sensitivity of AIME accuracy to logit noise injected at low- vs. high-entropy token positions across LRMs. (f) Tail-mass ratio and top-1 mismatch rate across token-entropy bins.

## 2.2. Large Reasoning Models

**Quantized reasoning models.** Quantized reasoning models have primarily been developed using PTQ, QAT and QAD. Several FP4 reasoning models are obtained via PTQ using vendor-supported pipelines (Nvidia, 2025; NVIDIA, 2025c), often combined with QAD (NVIDIA, 2026) or mixed-precision strategies to mitigate accuracy loss (NVIDIA, 2025b). QAT has also been explored by applying supervised fine-tuning (SFT) under simulated quantization noise, optimizing the standard token-level negative log-likelihood $\mathcal{L}_{\text{SFT}}(\theta) = -\mathbb{E}_{(X,Y)}\left[\frac{1}{T}\sum_{t=1}^{T}\log P_\theta(y_t \mid y_{<t}, X)\right]$. However, prior work reports instability or limited effectiveness for LRMs under W4A4KV4 settings (Lv et al., 2026).

**Token-level entropy in LRMs.** Token-level entropy is defined as the Shannon entropy of the next-token distribution $p_t(\cdot)$ at decoding step $t$, $H_t = -\sum_{v\in\mathcal{V}} p_t(v) \log p_t(v)$. In LRMs, token-level entropy has been used as a diagnostic tool for analyzing the semantic structure of reasoning traces (Wang et al., 2025a). Low-entropy tokens typically correspond to confident and deterministic predictions (e.g., digits or operators), whereas high-entropy tokens mark transition points that influence the subsequent reasoning path and have been used as targets for training or analysis.

## 3. Empirical Analysis

In LRMs, most tokens exhibit low entropy and correspond to confident predictions, while a small fraction (approxi-

mately 20%) have high entropy and mark transition points in the reasoning process (Wang et al., 2025a). In this section, we explore how quantization noise affects high- and low-entropy tokens during decoding, to better understand the causes of FP4 reasoning failures. Additional details on visualization, analysis setup, and implementation are provided in the Appendix B.1.

### 3.1. Semantic Grounding of Token Entropy

We group over 1.5M generated tokens by their average predictive entropy. As shown in Fig. 2(a), low-entropy tokens are dominated by digits and symbolic operators, whereas high-entropy tokens are enriched for discourse markers and connective phrases. Fig. 2(b) further shows that quantization affects both types of tokens by reducing the probability of the top-1 token while increasing the probability of alternative tokens, *flattening* the predictive distribution and increasing entropy. This observation motivates two failure hypotheses. For low-entropy tokens, where the model is originally confident, quantization increases the chance of sampling a non-top-1 alternative, potentially introducing symbolic errors. For high-entropy tokens, where the model is already uncertain about the next token, quantization can *flip* the top-1 prediction to a different token. This leads to a key question: is quantization-induced reasoning degradation primarily driven by increased sampling errors at low-entropy token predictions, or by top-1 instability at inherently uncertain high-entropy transitions?

## 3.2. Quantization Sensitivity via Token Entropy

**Entropy-aware mixed-precision decoding.** We study how quantization noise on low- and high-entropy tokens affects reasoning accuracy. As illustrated in Fig. 2(c), we perform entropy-aware mixed-precision decoding, where each next-token prediction is dynamically routed to either a full-precision model (BF16) or a quantized model (FP4) based on the token's predictive entropy. At each decoding step, we compute the token entropy from BF16 model and apply one of four routing strategies: routing low-entropy tokens to BF16 or to FP4 model, and routing high-entropy tokens to BF16 or to FP4 model. This controlled intervention isolates which token predictions during reasoning are most sensitive to reduced precision. As shown in Fig. 2(d), routing *low-entropy* prediction to BF16 recovers a large fraction of the accuracy lost under quantization, whereas routing only *high-entropy* prediction to BF16 yields relatively small improvement. These results indicate that quantization-induced failures are dominated by errors at *low-entropy token prediction*, rather than high-entropy token prediction. We observe a similar trend on additional examples (Fig. 8 in Appendix), where routing high-entropy tokens to the BF16 model does not lead to measurable accuracy gains. However, routing-based mixed-precision decoding is expensive to evaluate auto-regressively at scale, as it requires token-level routing and dual forward passes at each decoding step. Therefore, we perform a more scalable analysis by selectively injecting noise into high- and low-entropy tokens.

**Logit-noise sensitivity.** To test whether quantization errors on low-entropy token predictions consistently lead to reasoning failures, we conduct a controlled logit-perturbation experiment and evaluate its impact on reasoning benchmark accuracy (details in Appendix B.2). Instead of switching between full-precision and quantized models during decoding, which is time-consuming, we inject noise directly into the selected logits as a proxy for quantization error. Let $Z$ denote the pre-softmax logits produced by the model at a given decoding step, we add element-wise multiplicative Gaussian noise $\sigma Z \odot \eta$ to the logits, where $\eta \sim \mathcal{N}(\mathbf{0}, I)$ and $\sigma$ controls the noise magnitude. At each batched decoding step, we inject noise into the logits of either the top 25% highest-entropy token predictions or the bottom 25% lowest-entropy token predictions, and compare their effects on AIME-2025 accuracy. As shown in Fig. 2(e), injecting noise on low-entropy token predictions causes a large drop in accuracy across models, while perturbations applied only to high-entropy token predictions have a much smaller effect. Qualitative examples in Fig. 9 in Appendix further indicate that corrupting low-entropy token prediction often introduces minor symbolic errors (e.g., incorrect digits or operators) yet such errors often cascade into complete reasoning failures. In contrast, even substantial perturbations confined to high-entropy predictions frequently preserve the final answer, despite noticeably altering the surface form of the generated explanation.

## 3.3. Low-Entropy Tokens Fail under FP4 Sampling

Low-entropy tokens correspond to confident predictions, and their top-1 token is expected to remain stable under quantization. However, FP4 can still lead to reasoning failures at such tokens. One hypothesis is that quantization increases the probability of sampling a different token for these confident predictions. To quantify this effect, we measure the increase in probability mass assigned to non-top-1 tokens under FP4. Specifically, we define the *tail-mass* as $M = 1 - P(x_{\text{top1}})$, and summarize the quantization-induced inflation using the tail-mass ratio $\rho = (M_{\text{FP4}} + \epsilon)/(M_{\text{BF16}} + \epsilon)$, where $\rho > 1$ indicates an increased chance of sampling a non-top-1 alternative despite an unchanged top-1 rank. We additionally report the *top-1 mismatch ratio*, defined as the fraction of token positions where the argmax token under FP4 differs from that under BF16. As shown in Fig. 2(f), in low-entropy tokens the top-1 mismatch ratio remains close to zero, indicating that the argmax token is typically preserved under FP4; however, the tail-mass increases substantially, reflecting a higher probability of sampling non-top-1 alternatives. This observation is aligned with prior work showing that *flattening* low-entropy token distributions degrades reasoning accuracy (Wang et al., 2025a).

These results indicate that existing PTQ and QAT methods—which do not account for quantization-induced increases in non-top-1 sampling probability at low-entropy token predictions—may struggle to fully recover reasoning accuracy under FP4.

## 4. Methods

Motivated by the observations in Sec. 3 that quantization-induced errors disproportionately affect low-entropy predictions, we propose ReQAT, a unified framework for training FP4 LRMs that explicitly addresses this failure mode.

ReQAT consists of three complementary components. First, **TAQ** (*Trace-Aligned Quantization-Aware Training*) is a two-stage QAT procedure, where a BF16 fine-tuned model is first obtained in Stage-1 and then further optimized with QAT in Stage-2. Crucially, Stage-2 QAT is performed on the same reasoning traces used during Stage-1, ensuring that quantization-aware updates repeatedly act on the same low-entropy token decisions. Second, **SEM** (*Selective Entropy Minimization*) introduces an auxiliary loss that selectively reduces predictive entropy at low-entropy token positions, reinforcing confidence where sampling errors are most harmful. Finally, **Q-FIT** (*Quantization-Friendly Initialization via Transformation*) addresses residual degra-

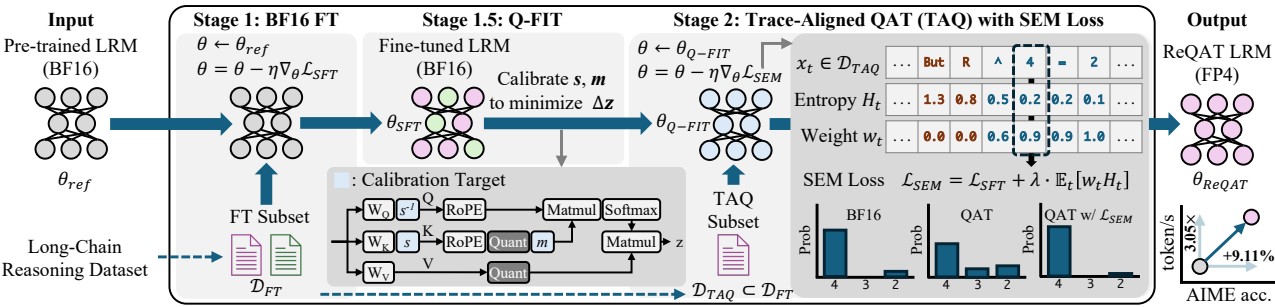

*Figure 3.* **ReQAT overview.** ReQAT performs BF16 FT, calibrates RoPE quantization parameters $(s, m)$ to minimize attention logit error $\Delta z$, and applies Q-FIT initialization followed by trace-aligned QAT with SEM loss on a TAQ subset to obtain an FP4 ReQAT model.

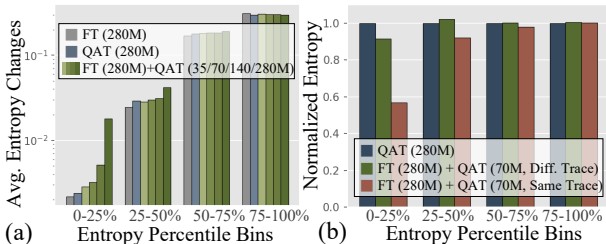

*Figure 4.* (a) Average entropy change across baseline entropy bins. (b) Normalized entropy across entropy bins.

dation from KV-cache quantization in W4A4KV4 by calibrating functionality-preserving transformations prior to Stage-2 QAT. Fig. 3 summarizes the overall pipeline.

## 4.1. TAQ: Trace-Aligned Quantization-Aware Training

TAQ is a two-stage QAT procedure that targets FP4 reasoning failures, which are dominated by sampling errors at low-entropy tokens (Sec. 3), by reusing the same reasoning traces across stages, ensuring that quantization-aware updates repeatedly act on the same high-confidence symbolic commitments under quantization.

**Motivation from entropy dynamics.** We analyze entropy dynamics during QAT using an entropy-change metric introduced in prior RL training work (Wang et al., 2025a), which measures how much token entropy changes from the base model during training. We consider three settings: (i) FT from base model with 280M tokens, (ii) QAT from base model with 280M tokens, and (iii) FT followed by an additional stage of QAT (FT+QAT). For FT+QAT, the second-stage QAT reuses the same training tokens as the first-stage FT, with the QAT budget varied from 35M to 280M tokens. As shown in Fig. 4(a), both FT and QAT primarily induce entropy changes in high-entropy bins, while low-entropy bins remain largely unchanged. In contrast, FT+QAT exhibits a different behavior: as Stage-2 QAT proceeds, entropy changes increasingly appear in low-entropy bins. This indicates that the two-stage procedure enables the

model to update low-entropy token predictions during QAT. Fig. 4(b) further compares the entropy magnitude across bins for QAT and FT+QAT. FT+QAT consistently results in lower entropy for low-entropy tokens than QAT. Importantly, this effect is not due to additional training alone, as it disappears when Stage-2 QAT is performed on different reasoning traces.

**Trace-aligned training.** TAQ implements a two-stage QAT procedure by constructing the Stage-2 QAT dataset from the *same reasoning traces* used during Stage-1 FT. As illustrated in Fig. 3, we first perform Stage-1 BF16 FT on a dataset $\mathcal{D}_{FT}$ to obtain a full-precision reasoning checkpoint. Stage-2 QAT is then applied to a subset $\mathcal{D}_{TAQ} \subseteq \mathcal{D}_{FT}$, ensuring that quantization-aware updates revisit the same reasoning traces and the same low-entropy token predictions encountered during FT. In practice, we find that a relatively small budget (e.g., 70M tokens) for $\mathcal{D}_{TAQ}$ is sufficient to reach accuracy comparable to BF16 fine-tuning, and this performance is maintained when the total fine-tuning budget is matched across methods.

## 4.2. SEM: Selective Entropy Minimization

Despite focusing updates on the same low-entropy token positions, trace alignment alone is not sufficient to fully recover FP4 reasoning accuracy, as illustrated in Fig. 1(c). Thus, recovery depends not only on *where* training is applied, but also on *how strongly* confidence at low-entropy positions is reinforced.

SEM strengthens low-entropy token predictions during TAQ by selectively minimizing predictive entropy. This entropy minimization is applied only to token positions expected to be near-deterministic, rather than uniformly across all tokens. As illustrated in Fig. 3, for each training sample with a reasoning trace $\{x_t\}$ from $\mathcal{D}_{TAQ}$, we compute the predictive entropy $H_t$ at each decoding step and assign a weight $w_t$ that controls the strength of entropy minimization. Tokens that are already near-deterministic, such as the digit "4", receive stronger entropy minimization via larger $w_t$.

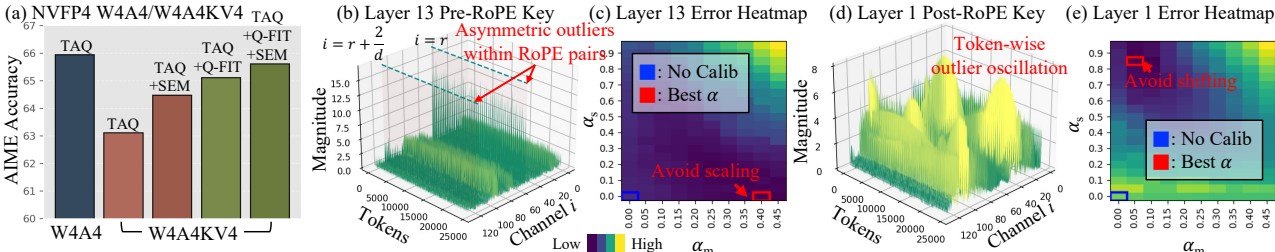

*Figure 5.* Visualization of Pre-/Post-RoPE key cache and calibration results via Q-FIT. More results are in Fig. 10 in Appendix.

Formally, SEM augments the standard SFT objective with an entropy minimization term:

$$\mathcal{L}_{\text{SEM}} = \mathcal{L}_{\text{SFT}} + \lambda \cdot \frac{1}{T} \sum_{t=1}^{T} w_t \, H_t, \qquad (1)$$

where $\lambda$ controls the strength of entropy sharpening and $w_t$ specifies where the regularizer is active. Rather than using a hard binary mask, we employ a soft weighting function parameterized by an entropy threshold $\tau$, preventing tokens near the threshold from being overly penalized:

$$w_t = \max\left(0, \ 1 - \frac{H_t - H_{\min}}{\tau - H_{\min} + \epsilon}\right),$$

where $H_{\min}$ is the minimum entropy within the minibatch, and $\epsilon$ ensures numerical stability. In practice, we set $\tau$ to the 75th percentile of the entropy values within each minibatch. Empirically, this soft weighting is more effective than a hard mask (Table 12). SEM differs from prior entropy minimization or regularization approaches in that it applies entropy control selectively based on token-level entropy, rather than uniformly across all tokens.

### 4.3. Q-FIT: Quantization-Friendly QAT Initialization

Quantizing the KV cache introduces additional challenges beyond W4A4 QAT. As shown in Fig. 5(a), TAQ alone is sufficient to largely recover accuracy under W4A4, but performance drops sharply under W4A4KV4. While SEM provides an additional ~1.3% accuracy gain, a substantial gap remains. Motivated by prior work on KV cache quantization for inference (Lin et al., 2025; Zhang et al., 2025b), we consider functionality-preserving transformations that reduce KV cache quantization error, including channel-wise pre-RoPE scaling and channel-wise post-RoPE shifting. However, using either transformation alone can be insufficient. As shown in Fig. 5(b), channels within the same RoPE pair can exhibit different outlier patterns, so a single shared scaling factor may not adequately reduce outliers across the pair. In addition, post-RoPE key magnitudes oscillate across tokens due to the rotational structure of RoPE (Hooper et al., 2024), making a fixed channel-wise offset potentially suboptimal over long decoding sequences (Fig. 5(d)).

Q-FIT is a lightweight initialization step that reduces KV cache quantization error *before* Stage-2 QAT. Instead of relying on a single transformation, Q-FIT jointly calibrates channel-wise scaling before RoPE and shifting after RoPE:

$$\tilde{Q} = \mathcal{R}(Q^{\text{pre}} \odot s), \qquad \tilde{K} = \mathcal{R}(K^{\text{pre}} \oslash s) - m. \quad (2)$$

The scaling vector $s$ is folded into the projection weights and introduces no inference-time overhead, while the shift vector $m$ is fixed after calibration and applied as a subtraction during inference. We parameterize $s$ and $m$ using two scalars $(\alpha_s, \alpha_m) \in [0, 1]$. For paired scaling, we first compute the maximum absolute value within each RoPE half, $s_0 = \max(\max |K_{0:d/2}|, \max |K_{d/2:d}|)$, and define $s = s_0^{\alpha_s}$, where $\alpha_s = 0$ disables scaling. For shifting, we initialize $m$ as the channel-wise mean of post-RoPE keys on a calibration set and scale it by $\alpha_m$, where $\alpha_m = 0$ disables shifting. Then, we select $(\alpha_s, \alpha_m)$ by minimizing the distance between BF16 and KV4 attention outputs. Q-FIT adapts the use of scaling and shifting to layer-specific Key cache characteristics. As shown in Fig. 5(b–c), when RoPE-paired channels exhibit asymmetric outliers and token-wise variation is relatively small, Q-FIT avoids paired scaling (i.e., $\alpha_s = 0$) and primarily relies on post-RoPE shifting. In contrast, as shown in Fig. 5(d–e), when key magnitudes exhibit strong token-wise oscillations, Q-FIT suppresses shifting and applies paired scaling instead. As a result, Q-FIT significantly improves W4A4KV4 QAT accuracy, as shown in Fig. 5(a).

**Additional implementation details.** For MXFP4 configurations, we apply a block-wise rotation prior to quantization, treated as a quantization-friendly special case within Q-FIT, following common MXFP4 practice (Castro et al., 2025; Tseng et al., 2025). For NVFP4, this step is unnecessary. Challenges in MXFP4 QAT are discussed in Appendix C.2. Additionally, we use the E1M2 FP4 format for the KV cache, as it consistently yields lower training loss (Fig. 10(b)).

## 5. Experiments

### 5.1. Experimental Settings

We evaluate three FP4 deployment settings: MXFP4 W4A16, MXFP4 W4A4, and NVFP4 W4A4KV4. For Re-

*Table 1.* AIME accuracy comparison of quantization methods on R1-Qwen-14B. ⋆ marks the best accuracy per bit-precision.

| Bit-Precision | Method | Total Fine-tuning Tokens | | | |
|---|---|---|---|---|---|
| | | **140M** | **210M** | **280M** | **350M** |
| BF16 | Baseline | 56.83 | | | |
| | FT | 63.70 | 64.17 | 65.46 | 64.79 |
| MXFP4 W4A16 | Direct PTQ | 50.37 | | | |
| | FT + PTQ | 54.38 | 57.71 | 56.87 | 56.77 |
| | QAT | 59.88 | 61.35 | 61.09 | 62.29 |
| | **ReQAT**$_T$ | 61.15 | 63.65 | 65.00 | 67.29 |
| | **ReQAT**$_{TQ}$ | 61.36 | 65.32 | 66.04 | 66.98 |
| | **ReQAT**$_{TQS}$ | **65.00** | **66.25** | **67.08** | **68.02**⋆ |
| MXFP4 W4A4 | Direct PTQ | 43.96 | | | |
| | FT + PTQ | 45.21 | 49.17 | 49.48 | 48.33 |
| | QAT | 54.59 | 55.67 | 54.06 | 58.03 |
| | **ReQAT**$_T$ | 56.15 | 59.79 | 59.48 | 59.69 |
| | **ReQAT**$_{TQ}$ | 59.48 | 62.60 | 64.27 | 64.48 |
| | **ReQAT**$_{TQS}$ | **59.69** | **62.81** | **64.48** | **65.94**⋆ |
| NVFP4 W4A4KV4 | Direct PTQ | 50.13 | | | |
| | FT + PTQ | 55.00 | 55.83 | 55.21 | 55.73 |
| | QAT | 57.09 | 57.60 | 58.86 | 58.23 |
| | **ReQAT**$_T$ | **60.32** | 60.42 | 63.13 | 63.12 |
| | **ReQAT**$_{TQ}$ | 59.79 | 63.44 | **65.94**⋆ | 65.21 |
| | **ReQAT**$_{TQS}$ | 59.79 | **64.28** | 64.37 | 65.63 |

*Table 2.* Results on R1-Llama-8B under NVFP4 W4A4KV4 (Total fine-tuning budget: 350M tokens) across various benchmarks.

| Bit-Precision | Method | GSM8K | MATH-500 | AIME-120 |
|---|---|---|---|---|
| BF16 | Baseline | 88.49 | 90.00 | 36.67 |
| | FT | 91.15 | 92.18 | 48.75 |
| NVFP4 W4A4KV4 | Direct PTQ | 86.45 | 84.62 | 23.13 |
| | FT + PTQ | 88.42 | 88.53 | 34.06 |
| | **ReQAT**$_T$ | 89.38 | 89.80 | 38.34 |
| | **ReQAT**$_{TQ}$ | **89.86** | **90.72** | 40.32 |
| | **ReQAT**$_{TQS}$ | 89.85 | 90.53 | **41.85** |

QAT, we report stage-wise variants, where **T**, **TQ**, and **TQS** denote TAQ only, TAQ with Q-FIT, and the full method with SEM, respectively; unless otherwise specified, **ReQAT** refers to the full **TQS** configuration. For TAQ, the total fine-tuning budget is split into BF16 FT and a fixed 70M-token QAT stage on $\mathcal{D}_{TAQ} \subset \mathcal{D}_{FT}$, with budgets matched across methods. We compare against BF16 FT, QAT, QAD and representative PTQ techniques, reporting the best accuracy under varying budgets. Experiments are conducted on R1-Qwen-14B and R1-Llama-8B (Guo et al., 2025). AIME-120 (2022–2025) is the primary metric, with GSM8K (Cobbe et al., 2021) and MATH-500 (Hendrycks et al., 2021) reported as complementary benchmarks. Additional details are provided in Appendix D.1.

## 5.2. Experimental Results

**Impact of ReQAT.** Table 1 shows AIME accuracy on R1-Qwen-14B as the total fine-tuning budget increases. Under BF16, fine-tuning yields large gains over the baseline but saturates around 280M tokens. QAT exhibits similar saturation: although it improves over PTQ, it remains well below BF16 FT even with larger budgets. In contrast, ReQAT continues

*Table 3.* Comparison with existing PTQ methods on AIME benchmark across different quantization settings (R1-Qwen-14B).

| Bit-Precision | FT | Method | Format | Group | Acc. |
|---|---|---|---|---|---|
| BF16 | – | Baseline | BF16 | – | 56.83 |
| | ✓ | FT | BF16 | – | 65.46 |
| W4A16 | – | AWQ | INT4 | 128 | 53.02 |
| | – | AWQ | MXFP4 | 32 | 51.46 |
| | ✓ | FT + AWQ | INT4 | 128 | 63.33 |
| | ✓ | FT + AWQ | MXFP4 | 32 | 60.11 |
| | ✓ | QAD | MXFP4 | 32 | 54.29 |
| | ✓ | QAT | MXFP4 | 32 | 62.29 |
| | ✓ | **ReQAT** | MXFP4 | 32 | 68.02 |
| | ✓ | **ReQAT** | NVFP4 | 16 | **68.75** |
| W4A4KV4 | – | QuaRot | INT4 | Row | 40.42 |
| | – | FlatQuant | INT4 | Row | 49.16 |
| | ✓ | FT + QuaRot | INT4 | Row | 47.92 |
| | ✓ | FT + FlatQuant | INT4 | Row | 56.15 |
| | ✓ | QAT | NVFP4 | 16 | 58.86 |
| | ✓ | **ReQAT** | NVFP4 | 16 | **65.63** |

to improve with additional budget and consistently matches or exceeds the best BF16 FT accuracy across all FP4 settings. For example, under NVFP4 W4A4KV4, ReQAT reaches 65.94% AIME accuracy, surpassing both the BF16 baseline (56.83%) and BF16 FT (65.46%) without increasing the total training budget. Accuracy increases monotonically as TAQ, Q-FIT, and SEM are added. Table 2 reports results on R1-Llama-8B under NVFP4 W4A4KV4 with a fixed 350M-token budget. ReQAT consistently outperforms PTQ baselines across benchmarks. On GSM8K and MATH-500, TAQ and Q-FIT recover most of the accuracy loss, while SEM provides only marginal changes. In contrast, on the more challenging AIME benchmark, SEM yields a clear gain, improving accuracy from 40.32% to 41.85%. This suggests that reinforcing low-entropy token confidence is particularly effective for harder reasoning tasks.

**Comparison against other methods.** Table 3 compares ReQAT with representative PTQ- and QAT-based methods on AIME under different quantization settings. Recent PTQ methods such as AWQ, QuaRot, and FlatQuant recover a substantial portion of accuracy, but still fall below BF16 baseline or BF16 full fine-tuning. QAT and QAD further improve over PTQ, yet their gains remain limited even with fine-tuning, resulting in accuracy comparable to or only slightly above strong PTQ baselines. In contrast, ReQAT achieves the highest accuracy across all evaluated settings. Across both W4A16 and the more challenging W4A4KV4 configurations, ReQAT consistently outperforms existing PTQ and QAT approaches by a clear margin.

## 5.3. Throughput Evaluation

We evaluate end-to-end inference throughput using `trtllm-bench` (NVIDIA, 2024) on two NVIDIA Blackwell platforms, DGX Spark and B200. To reflect real-world batched serving scenarios with bursty concurrency (Wang

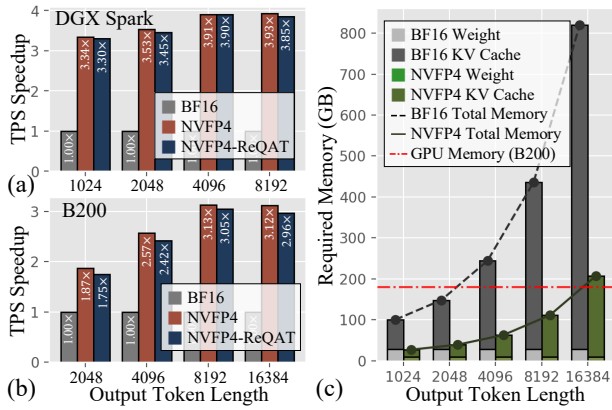

*Figure 6.* End-to-end throughput speedup for (a) DGX Spark and (b) B200. (c) BF16 vs. NVFP4 inference memory breakdown.

*Table 4.* Effect of two-stage QAT and trace alignment under different token budgets (MXFP4 W4A16 R1-Qwen-14B).

| Method | Trace Aligned | 140M | 210M | 280M | 350M |
|--------|---------------|------|------|------|------|
| QAT | – | 59.88 | 61.35 | 61.09 | 62.29 |
| FT+QAT | – | 60.10 | 59.89 | 62.19 | 62.60 |
| FT+QAT | ✓ | 61.15 | 63.65 | 65.00 | **67.29** |

*Table 5.* Ablation of Q-FIT designs (W4A4KV4 R1-Qwen-14B).

| Method | E1M2 KV | Pre-RoPE Scale | Post-RoPE Shift | Final Loss | AIME Acc. |
|--------|---------|----------------|-----------------|------------|-----------|
| ReQAT$_T$ | – | – | – | 0.7666 | 63.13 |
| ReQAT$_{TQ}$ | ✓ | ✓ | – | 0.7648 | 63.44 |
|  | ✓ | – | ✓ | 0.7634 | 62.71 |
|  | – | ✓ | ✓ | 0.7643 | 62.40 |
|  | ✓ | ✓ | ✓ | **0.7633** | **65.94** |

et al., 2025b), we evaluate throughput with 1K requests (512-token prompts), batching up to 256 requests, and maximum generation lengths of 8K/16K tokens on DGX Spark/B200. We report output-token throughput (TPS; tokens/s) relative to a BF16 baseline. Additional experimental details are provided in Appendix D.3. As shown in Fig. 6(a-b), NVFP4 achieves up to 3.93× (DGX Spark) and 3.13× (B200) throughput speedup over BF16. The overhead introduced by Q-FIT in ReQAT is small (4–5% compared to native NVFP4), while still delivering up to 3.90× and 3.05× speedup. We observe that NVFP4's throughput gains arise from two complementary factors: (i) larger batch sizes enabled by reduced weight and KV cache memory footprints, and (ii) improved compute efficiency from 4-bit GEMM enabled by activation quantization. Fig. 6(c) reports the memory breakdown at batch size 256 for BF16 and NVFP4 inference. It shows that BF16 cannot sustain large batch sizes at longer output lengths due to KV cache growth, whereas NVFP4's smaller memory footprint maintains a batch size of 256 up to 8K tokens. The TPS speedup correspondingly increases. At 16K output tokens, NVFP4 also becomes capacity-constrained, leading to speedup saturation (Fig. 6(b)). When both BF16 and NVFP4 fit within memory—i.e., at 2K output tokens on B200—there is no batch-size advantage; the observed 1.8–1.9× speedup is thus likely driven primarily by the compute benefit of 4-bit GEMM. On DGX Spark, NVFP4 already achieves over 3× TPS speedup even at short output lengths; for instance, it reaches 3.3× at 1K output tokens in Fig. 6(a), where the gain is largely consistent with the compute benefit.

### 5.4. Ablation Studies

We ablate SEM design choices, multi-epoch training with TAQ, decoding effects, comparison of Q-FIT with random rotation, and differences between reasoning and non-reasoning benchmarks in Appendix D.4.

**Role of trace alignment in TAQ.** Table 4 isolates the effect of trace alignment in two-stage QAT. Two-stage QAT without trace alignment yields limited accuracy gains (≤1%), whereas trace-aligned two-stage QAT improves accuracy about 5%. This supports that Stage-2 QAT compute becomes most effective when optimization repeatedly revisits the same token traces.

**Q-FIT designs.** Table 5 ablates Q-FIT components under NVFP4 W4A4KV4. Neither pre-RoPE scaling nor post-RoPE shifting alone provides reliable gains across budgets. The full Q-FIT design achieves the strongest accuracy, supporting that robust W4A4KV4 training benefits from jointly calibrating key transformations rather than committing to a single stabilization method.

**Robustness across response lengths.** We further analyze accuracy as a function of generated response length. This directly probes whether quantization errors accumulate over long reasoning traces. As shown in Table 6, PTQ suffers substantial degradation on longer responses, while ReQAT consistently maintains stronger accuracy among FP4 methods, achieving the best accuracy in the 24K–32K regime.

**Generalization to code generation.** To evaluate whether ReQAT generalizes beyond mathematical reasoning, we additionally evaluate on LiveCodeBench (Jain et al., 2024). Although ReQAT is trained using mathematical reasoning traces, it consistently improves over PTQ and QAT on code generation. As shown in Table 7, ReQAT$_{TQS}$ matches or slightly exceeds BF16 full fine-tuning while operating under FP4 quantization.

## 6. Discussions

An interesting question is how TAQ improves low-entropy token predictions during QAT (Fig. 4). Our observations suggest that repeatedly revisiting aligned reasoning traces may progressively concentrate learning signals on quantization-sensitive low-entropy token positions.

*Table 6.* Accuracy across response-length ranges (AIME-120, W4A4KV4 R1-Qwen-14B). Each cell reports `acc/#samples`.

| Response Length | 0–8K | 8–16K | 16–24K | 24–32K |
|---|---|---|---|---|
| BF16 Baseline | 92.1/355 | 55.1/303 | 21.9/178 | 12.9/124 |
| BF16 FT | 99.4/166 | 91.5/318 | 57.4/230 | 17.5/246 |
| PTQ | 84.6/363 | 48.5/297 | 14.7/204 | 6.3/96 |
| FT+PTQ | 98.8/170 | 80.5/302 | 42.3/213 | 12.7/275 |
| QAT | **100.0**/155 | 88.5/288 | 50.9/222 | 14.2/295 |
| **ReQAT** | 98.9/159 | **91.7**/333 | **55.7**/219 | **19.3**/249 |

*Table 7.* LiveCodeBench accuracy on R1-Qwen-14B.

| Method | MXFP4 W4A16 | NVFP4 W4A4KV4 |
|---|---|---|
| BF16 Baseline | 51.68 | 51.68 |
| BF16 FT | 53.68 | 53.68 |
| PTQ | 48.88 | 47.99 |
| QAT | 52.01 | 50.68 |
| **ReQAT**$_{\text{TQ}}$ | 53.82 | 53.08 |
| **ReQAT**$_{\text{TQS}}$ | **54.52** | **53.59** |

**Mechanism of TAQ.** To analyze the learning dynamics of TAQ, we study token-level embedding gradients. Let $E \in \mathbb{R}^{T \times D}$ denote the token embedding matrix and $G = \nabla_E \mathcal{L}$. We define the per-token gradient magnitude $s_t$ and the gradient contribution ratio of low-entropy tokens $C_{\text{low}}$:

$$s_t = \|G_{t,:}\|_2^2, \quad C_{\text{low}} = \frac{\sum_{t \in \mathcal{I}_{\text{low}}} s_t}{\sum_{t \in \mathcal{I}} s_t},$$

where $\mathcal{I}$ denotes all token positions and $\mathcal{I}_{\text{low}}$ denotes the low-entropy subset.

As shown in Fig. 7, revisiting aligned traces during QAT consistently increases $C_{\text{low}}$. Because the model has already learned the overall reasoning structure during Stage-1 BF16 fine-tuning, Stage-2 QAT allocates a larger fraction of its learning signal to correcting low-entropy token predictions that are vulnerable under quantization. In contrast, when QAT is performed on misaligned traces, this effect becomes substantially weaker, indicating diluted learning signals on quantization-sensitive token positions. These observations suggest that TAQ does not primarily rely on unusually large or specialized reasoning datasets. Instead, its main requirement is consistent supervision across stages, which enables gradient reallocation toward low-entropy token predictions during QAT.

**Limitations.** TAQ is currently built on top of SFT and therefore depends on the quality of the supervision signal provided by the SFT dataset. When the reasoning traces themselves are weak or noisy, TAQ may provide limited gains (Burns et al., 2024). Nevertheless, the underlying mechanism of TAQ—reinforcing quantization-sensitive low-entropy token predictions under quantization noise—may also be applicable to other supervision paradigms such as knowledge distillation, which we leave for future work.

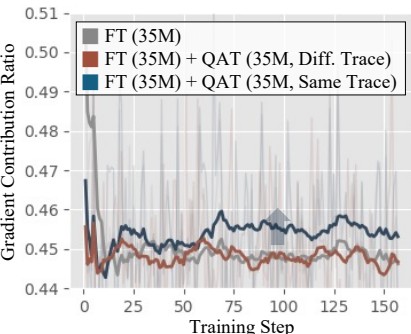

*Figure 7.* Gradient contribution ratio of low-entropy tokens during QAT. TAQ increases gradient allocation toward vulnerable low-entropy tokens, while misaligned traces dilute this effect.

## 7. Related Works

**FP4 quantization for LRMs.** Recent hardware platforms provide native support for microscaled FP4 formats such as MXFP4 and NVFP4 (Nvidia, 2024; AMD, 2025), enabling W4A16, W4A4, and W4A4KV4 deployments (OpenAI et al., 2025; Rouhani et al., 2023). However, substantial reasoning degradation under aggressive FP4 quantization has been widely reported for LRMs (Liu et al., 2025a; Li et al., 2025; Zhang et al., 2025c). Existing deployments rely on PTQ or mixed-precision designs (Nvidia, 2025; NVIDIA, 2025c;b;a), and prior QAT approaches remain limited under W4A4KV4 settings (Lv et al., 2026).

**Entropy-based analysis for LRMs.** Token-level entropy has been used as a diagnostic signal to analyze reasoning traces and training dynamics in LRMs (Wang et al., 2025a). It has also been leveraged in reinforcement learning and instruction-tuned models to modulate or study training behavior, such as controlling exploration or encouraging sharper predictions (Wang et al., 2025c; Agarwal et al., 2025), primarily in settings with limited reasoning capability. Recent work further explores the entropy-increasing effect of quantization noise as an analysis and control signal in RL-based reasoning pipelines (Huang et al., 2025). Unlike prior entropy-based studies that mainly use entropy as an analysis or exploration signal, our work identifies low-entropy token predictions as the primary failure point under FP4 quantization and directly targets them during QAT.

## 8. Conclusion

We introduce ReQAT, a reasoning-centric FP4 training framework for W4A4KV4 deployment of large reasoning models. By targeting low-entropy token failures, ReQAT matches or surpasses BF16 fine-tuning accuracy under the same training budget, while achieving up to $3.9\times$ throughput speedup on production hardware. These results demonstrate that FP4 quantization can simultaneously deliver high efficiency and strong reasoning performance.

## Acknowledgements

This work was supported by the National Research Foundation of Korea (NRF) grants funded by the Korea government (MSIT) (No. RS-2025-00561961 and No. RS-2023-00260527). This work was partly supported by the Institute of Information & Communications Technology Planning & Evaluation (IITP) grant funded by the Korea government (MSIT) (No. RS-2020-II201373, Artificial Intelligence Graduate School Program (Hanyang University)). This research was also supported by the High-Performance Computing Support Project and the Advanced GPU Utilization Support Program, funded by the Government of the Republic of Korea (Ministry of Science and ICT).

## Impact Statement

This paper presents work whose goal is to advance the field of machine learning. There are many potential societal consequences of our work, none of which we feel must be specifically highlighted here.

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

## A. Microscaling Formats

| Name | Element Data Type | Block Scale Data Type | Global Scale Data Type | Block Size |
|------|-------------------|----------------------|------------------------|------------|
| MXFP4 | FP4 (E2M1) | FP8 (E8M0) | - | 32 |
| NVFP4 | FP4 (E2M1) | FP8 (E4M3) | FP32 | 16 |

*Table 8.* Quantization configurations of MXFP4 and NVFP4.

Table 8 summarizes the quantization metadata of two widely used microscaled FP4 formats, MXFP4 and NVFP4. Both formats store elements in FP4 with an E2M1 layout, but differ in how scaling is represented and applied. In microscaled formats, values are typically interpreted with a block-wise scaling factor (shared across a fixed number of elements), optionally combined with an additional global scaling factor.

**MXFP4.** MXFP4 uses a block-wise scale stored in FP8 (E8M0) with block size 32, and does not use an explicit global scale. The larger block size amortizes scale overhead, while the E8M0 exponent-only scale provides a wide dynamic range for coarse magnitude adjustment. This design is commonly adopted in FP4 deployment regimes such as W4A16 and W4A4, where activations may remain in higher precision or where the KV cache is not aggressively quantized.

**NVFP4.** NVFP4 uses a smaller block size (16) with a block-wise FP8 scale in E4M3, and additionally applies a global FP32 scale. The extra global scale can help stabilize overall magnitude calibration, while the smaller block improves local adaptability at the cost of slightly higher metadata overhead. NVFP4 can be used in end-to-end low-precision settings (e.g., W4A4KV4) that also quantize the KV cache, where tighter control of scaling is beneficial for long-context decoding and KV cache growth.

## B. Details for Token Entropy Analysis in Sec. 3

### B.1. Token Entropy Analysis Settings (Fig. 2)

**Sampling setup.** Unless otherwise stated, all results are obtained using stochastic decoding with temperature $0.6$ and top-$p$ sampling with $p = 0.95$.

**Visualization of low- and high-entropy tokens (Fig. 2(a)).** For the word-cloud visualizations in Fig. 2(a), we collect token-level entropy statistics from long reasoning traces generated by R1-Qwen-14B on 90 prompts from AIME 2022–2024 (Maxwell-Jia, 2024). In total, more than 1.5M tokens and their corresponding entropies are extracted. We focus on frequently generated tokens (appearing more than 100 times) and visualize them by separating low-entropy and high-entropy tokens.

**Setting for entropy-aware mixed-precision decoding (Fig. 2(c-d)).** To analyze whether quantization errors are more critical when selecting low- or high-entropy tokens, we perform entropy-aware mixed-precision inference using two models: BF16 and MXFP4 W4A16 versions of R1-Qwen-14B. Following (Wang et al., 2025a), we set the entropy threshold $\tau = 0.6$, which corresponds to approximately the top 25% highest-entropy tokens based on the offline statistics collected from the 1.5M-token corpus.

At each decoding step, we run `model.forward()` from HuggingFace Transformers for both the BF16 and quantized models. The entropy of the BF16 model's predicted distribution is used to decide whether to select the next token from the BF16 model or the quantized model. The selected token is then fed back as input to both models in the subsequent decoding step. Fig. 2(c) presents a representative challenging example in which the BF16 model produces the correct answer in 6 out of 8 trials, while the quantized model succeeds only once. Additional examples are shown in Fig. 8, exhibiting similar behavior: allowing the quantized model to select only high-entropy tokens results in higher accuracy than allowing it to select only low-entropy tokens.

**Tail mass ratio and Top-1 mismatch ratio (Fig. 2(f)).** To investigate the impact of quantization noise on the model's output distribution, we analyze the redistribution of probability mass, specifically focusing on the relationship between the *Top-1 mismatch rate* and the *Tail mass ratio* across different entropy levels. We use the AIME (2022-2024) dataset and collect the full probability distributions from the R1-Qwen-14B model in both BF16 (reference) and NVFP4 (W4A4KV4) settings. For each token position $t$, we compute the entropy $H(p_t)$ of the BF16 distribution to represent the model's confidence. The data is then partitioned into $N = 20$ equal-sized bins based on these entropy values (quantiles).

Baseline vs. Quantized Model (Other Sample)

| Prompt (AIME-90 Sample 25) | | In isosceles trapezoid $ABCD$, parallel bases $\overline{AB}$ and $\overline{CD}$ have lengths $500$ and $650$, respectively, and $AD=BC=333$. The angle bisectors of $\angle{A}$ and $\angle{D}$ meet at $P$, and the angle bisectors of $\angle{B}$ and $\angle{C}$ meet at $Q$. Find $PQ$. | | | | | | | |
|---|---|---|---|---|---|---|---|---|---|
| Gold Answer | | 242 | | | | | | | |
| $H_t \geq \tau$ | $H_t < \tau$ | Seed | | | | | | | |
| | | 1 | 2 | 3 | 4 | 5 | 6 | 7 | 8 |
| Baseline Model | Baseline Model | - | 242 | 242 | - | 242 | 242 | 242 | 242 |
| Quantized Model | Baseline Model | 242 | 242 | 242 | 242 | - | - | 65 | - |
| Baseline Model | Quantized Model | 92 | - | - | 92 | - | 92 | - | 242 |
| Quantized Model | Quantized Model | 242 | - | 92 | 242 | - | 92 | 242 | - |

*Figure 8.* Example of entropy-aware mixed-precision decoding. We route low-entropy tokens to the full-precision model and high-entropy tokens to the quantized model, using an entropy threshold $\tau$ (e.g., $\tau = 0.6$).

## B.2. Controlled Logit-Noise Perturbation (Fig. 2(e))

**Setup.** Routing-based mixed-precision decoding (Fig. 2(c)) is expensive to evaluate auto-regressively at scale when comparing quantized and full-precision models during generation. To obtain a lightweight proxy, we inject logit perturbations into a fixed fraction of token decisions within each batched decoding invocation, while varying *which* decisions (high- vs. low-entropy) are perturbed.

**Entropy-based selection with fixed budget.** Let $Z \in \mathbb{R}^{N \times V}$ denote the pre-softmax logits from a single model invocation, where $N$ indexes token decisions processed in the current batched step (flattened across requests) and $V$ is the vocabulary size. For each decision $n \in \{1, \ldots, N\}$, we compute predictive entropy $H_n$ from $\mathrm{softmax}(Z_n)$. We then form low- and high-entropy subsets by selecting the bottom and top $q$ quantiles of $\{H_n\}_{n=1}^N$, respectively (with $q = 25\%$ unless otherwise stated). This quantile-based selection keeps the perturbation *budget* constant (i.e., a fixed fraction $q$ per invocation), enabling a fair comparison between perturbing low-entropy and high-entropy tokens.[2]

**Scale-aware logit perturbation.** We inject multiplicative, scale-aware Gaussian noise into the logits at the selected positions:

$$\tilde{Z} \;=\; Z \;+\; M \odot \big(\sigma\, Z \odot \eta\big), \tag{3}$$

where $\eta \sim \mathcal{N}(\mathbf{0}, I_{N \times V})$ is i.i.d. Gaussian noise, $\odot$ denotes element-wise multiplication, $\sigma$ controls noise magnitude, and $M \in \{0,1\}^{N \times 1}$ is a broadcastable mask selecting which token decisions are perturbed. We use this perturbation as a simple proxy for relative logit distortions induced by low-bit quantization.

**Observation.** Across model sizes and architectures, perturbing low-entropy decisions consistently causes a sharp drop in AIME-2025 accuracy, whereas perturbations restricted to high-entropy decisions have a substantially smaller impact and can occasionally be benign (Fig. 2(e)). Qualitative examples (Fig. 9) suggest that corrupting low-entropy decisions often introduces small symbolic errors (e.g., incorrect digits or operators) that cascade into complete reasoning failure, while perturbations confined to high-entropy decisions frequently preserve the final answer despite changing the surface form of the explanation.

## B.3. Qualitative Analysis of Entropy-Selective Logit Noise (Fig. 9)

To complement our quantitative analysis, we present a qualitative case study that illustrates the asymmetric impact of logit noise on low-entropy and high-entropy tokens during long-chain reasoning.

**Noise on high-entropy tokens.** When noise is injected into high-entropy tokens, the generated reasoning trace exhibits noticeable stylistic and lexical variation. For example, discourse markers and transition phrases (e.g., "Wow", "Hmm", or alternative connective expressions) appear, and the overall wording of the reasoning chain may differ substantially from the baseline. In some cases, entirely different transition tokens are sampled, leading to a reasoning trace that is superficially very different from the original one.

---

[2]Because selection is performed over the flattened set of decisions in the current batched step, individual requests may receive different amounts of perturbation depending on their local entropy profiles; however, the *global* perturbation rate is fixed by construction.

---

**<R1-Qwen-7B>**

Alright, so I've got this geometry problem here about a right triangle ABC with a right angle at A, and the hypotenuse BC is 38 units long. Inside this triangle, there are two points, K and L, such that the lengths AK, AL, BK, CL, and KL are all 14 units each. The goal is to find the area of quadrilateral BKLC, which can be expressed as n\u221a3, and then determine the integer n.\n\nFirst off, since ABC is a right-angled triangle at A, I can recall that in such triangles, the Pythagorean theorem holds. So, AB\u00b2 + AC\u00b2 = BC\u00b2. But BC is given as 38, so AB\u00b2 + AC\u00b2 = 38\u00b2 = 1444.\n\nNow, the points K and L are inside the triangle with all those equal lengths. Let me visualize this. Points K and L are such that AK = AL = BK = CL = KL = 14. Hmm, that's quite a configuration. So, K is 14 units away from A, B, and L. Similarly, L is 14 units away from A, C, and K.\n\nWait, that seems a bit confusing. Let me break it down. AK = 14, AL = 14, BK = 14, CL = 14, and KL = 14. So, K is a point inside the triangle that is 14 units away from A, B, and L. Similarly, L is 14 units away from A, C, and K. So, K and L are each equidistant from two vertices and from each other.\n\nThis makes me think that K and L might lie on the angle bisectors or maybe on some perpendicular bisectors? But since the triangle is right-angled, maybe they lie on the medians or something else. Hmm.\n\nAlternatively, since K is 14 units from A and B, it must lie somewhere on the intersection of two circles: one centered at A with radius 14 and another centered at B with radius 14. ...

**Answer: boxed{104} ✓**

---

**<R1-Qwen-7B with Noise on 25% High-Entropy Tokens (alpha=0.2)>**

Okay, so I've got this problem about a right triangle ABC with a right angle at A, and the length of BC is 38. There are some points K and L inside the triangle such that AK, AL, BK, CL, and KL are all 14 units long. The task is to find the area of quadrilateral BKLC, which can be expressed as n\u221a3, and determine the value of n.\n\nHmm, right triangles have nice properties, especially concerning the geometric mean. Let me sketch out what I know so far:\n\nFirst, it's a right triangle at A, so triangle ABC is right-angled at A, implies that AB and AC are the legs, and BC is the hypotenuse, which is given as 38. So, using Pythagoras, AB\u00b2 + AC\u00b2 = BC\u00b2 = 38\u00b2 = 1444.\n\nNow, there are points K and L inside the triangle such that AK = AL = BK = CL = KL = 14. That's quite a lot of equal lengths, so they must form some kind of equilateral component perhaps?\n\nWait, from the question, the quadrilateral BKLC is being considered. So since BK, CL, and KL are all 14, triangle BKL and triangle CKL might be equilateral? Hmm, not necessarily, not yet sure.\n\nAlternatively, because of the various equal lengths, maybe B, K, L, C form a rhombus or something symmetrical?\n\nAdditionally, points K and L are inside the triangle. Okay, so they're somewhere between the sides AB, BC, and AC.\n\nAlso, AK = 14 is mentioned, so point K is somewhere on the triangle such that its distance from A is 14. Similarly, AL = 14, so L is another point at distance 14 from A.\n\nSimilarly, BK = 14, so K is 14 units away from B as well. And CL = 14, so L is 14 units away from C.\n\nWow, so that makes two circles: ...

**Answer: boxed{104} ✓**

---

**<R1-Qwen-7B with Noise on 25% Low-Entropy Tokens (alpha=0.2)>**

Alright, so I've got this geometry problem here about a right triangle ABC with a right angle at A, and the hypotenuse BC is 38 units long. Inside this triangle, there are two points, K and L, such that the lengths AK, AL, BK, CL all equal 14 units, and also the distance between K and L is 14 units. The goal is to find the area of quadrilateralBKLC, which can be expressed as n\u221a3, and we need to find n.\n\nHmm, okay. Let me try to visualize this triangle first. Since ABC is a right-angled triangle at A, sides AB and AC are the legs (the ones forming the right angle), whereas hypotenuse BC is opposite the right angle. The length of BC is 38. So, if I denote AB as 'a' and AC as' b', then by Pythagoras theorem, a\u00b2 + b\u00b2 = 38\u00b2 = 144. Hmm, wait, 38\u00b2 is actually 1444, not 144. I think I made a mistake there. Let me correct that. 38 squared is indeed 1444, ...

**Answer: boxed{70} ✗**

---

*Figure 9.* Qualitative examples of entropy-selective logit-noise perturbation. Injecting noise into low-entropy token predictions often introduces minor symbolic errors that cascade into incorrect final answers, whereas perturbations applied only to high-entropy token predictions typically preserve the final answer despite altering the surface form of the reasoning trace.

Importantly, despite these variations, the core logical structure of the solution remains intact. The model preserves correct arithmetic operations, symbolic relationships, and geometric constraints, and successfully arrives at the correct final answer. This suggests that high-entropy regions primarily govern narrative flow and exploration, and are therefore tolerant to significant perturbations without compromising correctness.

**Noise on low-entropy tokens.** In contrast, injecting noise into low-entropy tokens leads to immediate and catastrophic failure. Even small perturbations frequently cause errors in simple arithmetic (e.g., incorrect squaring or summation) or the mis-selection of critical symbols such as numerical constants. Once such an error occurs, the subsequent reasoning becomes increasingly incoherent, as later steps are built upon an incorrect intermediate result.

Qualitatively, we observe that the reasoning trace quickly diverges from any valid solution path, exhibiting confusion, contradiction, or premature termination. In all tested cases, the model fails to recover and does not reach the correct answer when low-entropy anchors are corrupted.

## C. ReQAT Details

### C.1. Multi-Epoch Training vs. Quantization

A natural question is whether FP4 brittleness can be resolved simply by training longer on fixed reasoning traces. Table 9 shows that increasing FT epochs before PTQ does not close the NVFP4 W4A4KV4 gap and can even widen it (up to $15.42$ points at $k = 4$). Likewise, extending QAT from a fixed $\text{FT}_{4\text{-}ep}$ checkpoint yields limited improvement, with the drop remaining around 12–13 points. These results suggest that recovery is not driven by more iterations alone, but by *where* the learning signal is applied—motivating SEM to reinforce low-entropy anchors (Section 4.2).

*Table 9.* Accuracy drop $\Delta$ (AIME-120) under NVFP4 W4A4KV4 on R1-Llama-8B, measured relative to the corresponding full-precision FT baseline.

| FT/QAT Epochs ($k$) | 1 | 2 | 3 | 4 | 5 |
|---|---|---|---|---|---|
| FT ($k$-ep) $\rightarrow$ PTQ | 11.05 | 14.79 | 15.21 | 15.42 | 14.69 |
| FT$_{4\text{-}ep}$ $\rightarrow$ QAT ($k$-ep) | 13.02 | 12.09 | 12.50 | 11.88 | 13.65 |

## C.2. Detailed Algorithm and Design for Q-FIT

---
**Algorithm 1** Q-FIT: Calibration of Pre-RoPE Paired Scaling and Post-RoPE Key Shift
---
**Require:** Pre-RoPE $Q^{\text{pre}}, K^{\text{pre}}, V$; RoPE $(\cos, \sin)$; KV quantizer $Q_{\text{KV4}}(\cdot)$; #heads $H$, #KV heads $H_{\text{kv}}$, head dim $d$; grid sizes $G_s, G_m; \varepsilon$

**Ensure:** Paired scales $S_Q^\star, S_K^\star$ and post-RoPE shift $m^\star$

1: $n_{\text{rep}} \leftarrow H/H_{\text{kv}}$      ▷ GQA replication factor
2: $\bar{d} \leftarrow d/2$      ▷ RoPE pairing half-dimension
3: **(Init paired scale from key magnitudes)**
4: **for** $h = 1, \ldots, H_{\text{kv}}$ **do**
5:      **for** $r = 1, \ldots, \bar{d}$ **do**
6:          $M_{h,r}^{(1)} \leftarrow \max_t \left| K_{t,h,r}^{\text{pre}} \right|$
7:          $M_{h,r}^{(2)} \leftarrow \max_t \left| K_{t,h,r+\bar{d}}^{\text{pre}} \right|$
8:          $S_{K,h,r}^0 \leftarrow \max\left( \max(M_{h,r}^{(1)}, M_{h,r}^{(2)}), \varepsilon \right)$
9:          $S_{K,h,r+\bar{d}}^0 \leftarrow S_{K,h,r}^0$      ▷ enforce $s_{h,r} = s_{h,r+\bar{d}}$
10:      **end for**
11: **end for**
12: $S_Q^0 \leftarrow \text{Replicate}(S_K^0, n_{\text{rep}})$      ▷ map KV-head scale to query heads
13: **(Init post-RoPE shift template)**
14: $\tilde{K}^0 \leftarrow \text{RoPE}(K^{\text{pre}}; \cos, \sin)$
15: $\mu \leftarrow \mathbb{E}_t\left[ \tilde{K}_t^0 \right]$      ▷ channel-wise mean over tokens
16: **(BF16 reference attention)**
17: $\tilde{Q}^0 \leftarrow \text{RoPE}(Q^{\text{pre}}; \cos, \sin)$
18: $Y_{\text{ref}} \leftarrow \text{Attn}(\tilde{Q}^0, \tilde{K}^0, V)$
19: $E^\star \leftarrow +\infty$
20: **for** $a = 0, \ldots, G_s - 1$ **do**
21:      $\alpha_s \leftarrow a/G_s$
22:      $S_K(\alpha_s) \leftarrow (S_K^0)^{\alpha_s}$
23:      $S_Q(\alpha_s) \leftarrow (S_Q^0)^{\alpha_s}$
24:      **for** $b = 0, \ldots, G_m - 1$ **do**
25:          $\alpha_m \leftarrow b/G_m$
26:          $m(\alpha_m) \leftarrow \alpha_m \cdot \mu$
27:          **(Apply RoPE-consistent transformations)**
28:          $\tilde{Q} \leftarrow \text{RoPE}\left( Q^{\text{pre}} \odot S_Q(\alpha_s); \cos, \sin \right)$
29:          $\tilde{K} \leftarrow \text{RoPE}\left( K^{\text{pre}} \oslash S_K(\alpha_s); \cos, \sin \right)$
30:          $\hat{K} \leftarrow Q_{\text{KV4}}(\tilde{K} - m(\alpha_m))$      ▷ KV4 cache
31:          $\hat{V} \leftarrow Q_{\text{KV4}}(V)$      ▷ optional: match KV4 path
32:          $Y \leftarrow \text{Attn}(\tilde{Q}, \hat{K}, \hat{V})$
33:          $E \leftarrow \text{MSE}(Y_{\text{ref}}, Y)$
34:          **if** $E < E^\star$ **then**
35:              $E^\star \leftarrow E$
36:              $S_Q^\star \leftarrow S_Q(\alpha_s), \;\; S_K^\star \leftarrow S_K(\alpha_s), \;\; m^\star \leftarrow m(\alpha_m)$
37:          **end if**
38:      **end for**
39: **end for**
40: **return** $(S_Q^\star, S_K^\star, m^\star)$
---

**More layer-wise results.** Fig. 10(a) presents additional layer-wise calibration results. Across different layers, Q-FIT consistently reduces post-RoPE calibration errors.

**Training loss.** Fig. 10(b) shows the training loss under different QAT configurations. Compared to TAQ-only baselines, incorporating Q-FIT results in lower and more stable training loss, suggesting improved optimization behavior during

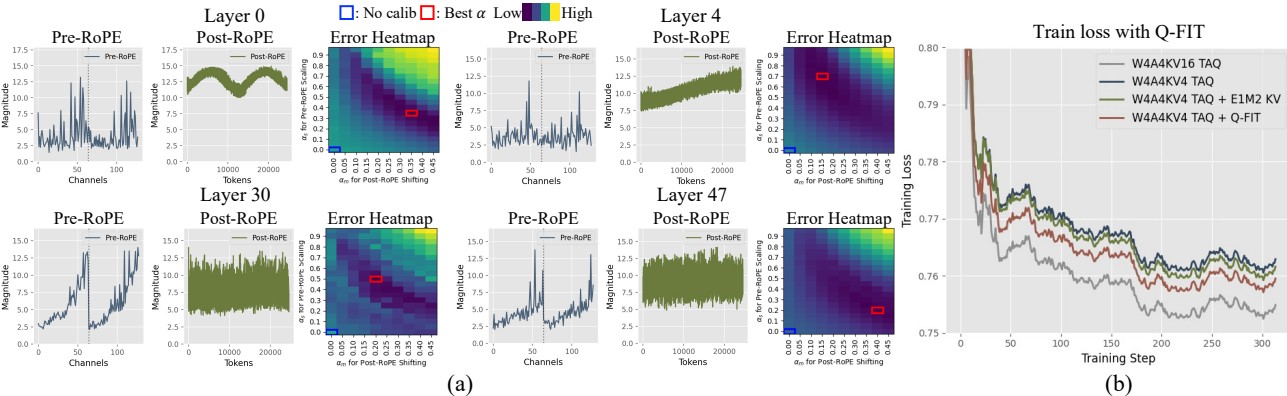

*Figure 10.* (a) Layer-wise calibration behavior with Q-FIT. We visualize key-cache statistics before and after RoPE across multiple layers, together with the corresponding calibration error heatmaps. (b) Training loss comparison under different QAT settings. Using an E1M2 KV cache consistently results in lower training loss compared to TAQ, which adopts E2M1 by default. Incorporating Q-FIT further reduces the training loss relative to TAQ-only variants, indicating more stable and effective quantized training (Model: R1-Qwen-14B).

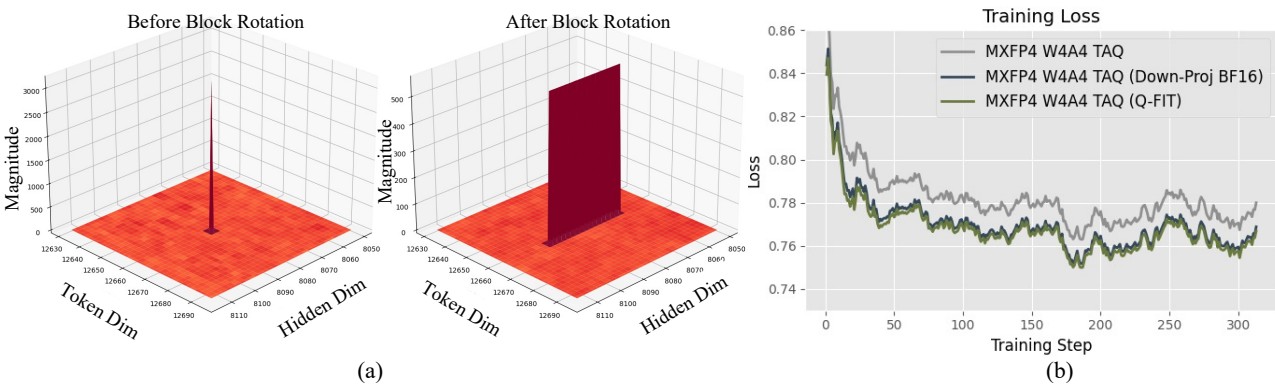

*Figure 11.* Effect of block rotation and Q-FIT in MXFP4 W4A4. (a) Activation magnitude before and after block rotation, showing a small number of extreme outlier channels. After block-wise Hadamard rotation, outliers are redistributed across channels. (b) Training loss comparison under different QAT settings. Applying block rotation with Q-FIT achieves a training loss comparable to leaving the MLP down-projection layer unquantized, and consistently improves over TAQ with fully quantized down-projection (Model: R1-Qwen-14B).

quantized fine-tuning.

**Challenges in MXFP4 activation quantization.** Large activation outliers are a well-known source of significant error in MXFP4 W4A4 during both training and inference (Tseng et al., 2025; Castro et al., 2025; Lee et al., 2025). A commonly adopted mitigation strategy during training is block rotation (Castro et al., 2025), which typically employs a randomized Hadamard transform (Ashkboos et al., 2024) to redistribute information within a block and reduce the impact of outliers. As illustrated in Fig. 11(a), block rotation effectively spreads extreme activation values across channels.

In training, the challenges posed by activation outliers are particularly pronounced in the MLP down-projection layers of Transformers due to the use of SwiGLU activations (Fishman et al., 2025). As shown in Fig. 11(b), excluding the down-projection layer from quantization leads to a substantial reduction in training loss. We incorporate block-wise Hadamard transforms into MXFP4 as part of Q-FIT to make the model more amenable to QAT. With this design, we observe training loss curves that are nearly identical to those obtained when leaving the down-projection layer unquantized. We note that such block rotation is not applied for NVFP4, which employs more fine-grained and precise scaling factors that sufficiently mitigate activation outliers without requiring additional rotations.

# D. Experimental Details and More Results

## D.1. Experimental Details

**Quantization settings.** We evaluate ReQAT under three practical FP4 deployment configurations: MXFP4 W4A16 (weight-only), MXFP4 W4A4 (weight–activation), and NVFP4 W4A4KV4 (end-to-end FP4 with a 4-bit KV cache). For all settings, every linear layer in the Transformer decoder is quantized.

We compare against representative PTQ baselines, including AWQ (Lin et al., 2024) for W4A16, and QuaRot (Ashkboos et al., 2024) and FlatQuant (Sun et al., 2025) for W4A4KV4. We additionally include a *FT+PTQ* baseline, which applies advanced PTQ after BF16 fine-tuning, to isolate the effect of post-training refinement from QAT.

For ReQAT, we use $\lambda = 0.1$ by default. The entropy threshold $\tau$ is set to the 75th percentile of token entropy, so that SEM is applied to the lowest-entropy 75% of tokens, while the highest-entropy 25% are left unaffected. For PTQ baselines, we disable the clipping option in AWQ under MXFP4 due to reduced accuracy, and apply GPTQ (Frantar et al., 2023) within QuaRot. For INT4 methods under W4A4KV4, *Row* in Table 3 denotes row-wise quantization for weights and activations, with the KV cache quantized using group size 128.

**Models and tasks.** Our main model is R1-Qwen-14B (Guo et al., 2025), and we additionally evaluate transfer on R1-Llama-8B (Guo et al., 2025). We focus on mathematical reasoning and use AIME-120 (Maxwell-Jia, 2024) accuracy (AIME 2022–2025) as the primary metric, aggregating four years of AIME to reduce variance from limited test samples. We also report results on MATH-500 (Hendrycks et al., 2021) and GSM8K (Cobbe et al., 2021) as complementary benchmarks. All results are averaged over 8 random seeds.

*Table 10.* Year-wise AIME accuracy on R1-Qwen-14B under NVFP4 W4A4KV4. For fine-tuned methods, we report the best accuracy across training budgets.

| Method | 2022 | 2023 | 2024 | 2025 | Avg. |
|---|---|---|---|---|---|
| BF16 Baseline (ours) | 56.67 | 52.92 | 68.33 | 49.38 | 56.83 |
| BF16 Baseline (official) | - | - | 69.70 | - | - |
| BF16 Baseline (reported in (Luo et al., 2026)) | - | - | - | - | 57.50 |
| BF16 Baseline (reported in (Liu et al., 2025a)) | - | - | - | - | 54.70 |
| BF16 FT | 66.25 | 64.58 | 72.08 | 58.96 | 65.47 |
| PTQ | 48.33 | 49.58 | 59.58 | 42.92 | 50.10 |
| FT+PTQ | 55.00 | 56.67 | 58.33 | 52.92 | 55.73 |
| QAT | 60.83 | 56.25 | 67.50 | 50.83 | 58.85 |
| **ReQAT** | **67.92** | **65.42** | **71.25** | **59.17** | **65.94** |

**Evaluation protocol.** Our main AIME results report average accuracy over AIME-120, covering AIME 2022–2025. This differs from evaluations that report results only on AIME-2024. Under the same AIME-2024 setting, our reproduced BF16 baseline achieves 68.33%, closely matching the official DeepSeek report of 69.70%. Table 10 provides the year-wise breakdown.

## D.2. Training Details

**Training details.** We train on the Math subset of OpenThought-3 (Guha et al., 2025), randomly sampling examples to match the target fine-tuning token budget. For R1-Qwen-14B, Stage-1 BF16 fine-tuning is performed for a single epoch while increasing the token budget. For R1-Llama-8B, we instead train on a fixed 70M-token subset with multiple epochs; for example, a 280M-token budget corresponds to four epochs over the same 70M-token subset. For TAQ, we fix the Stage-2 QAT budget to 70M tokens. For Q-FIT calibration, we use Wikitext-2 (Merity et al., 2016) with an input length of 512 and 256 samples. This calibration takes approximately 7 minutes on a single H200 GPU for R1-Llama-8B. Additional training details and hyperparameters are provided in Table 11.

## D.3. Throughput Evaluation Details

All throughput experiments are conducted on two NVIDIA Blackwell platforms: DGX Spark and a single B200 GPU. DGX Spark has compute capability `sm_121` with 128 GB unified memory and 283 GB/s memory bandwidth, delivering 500 TFLOPS for FP4 inference. In contrast, B200 has compute capability `sm_100a` with 180 GB HBM3e and 8 TB/s bandwidth, delivering 9 PFLOPS for FP4. We use TensorRT-LLM v1.2.0rc8 with CUDA 13.0, PyTorch 2.9.0, and Python 3.12.3. For

*Table 11.* Key hyperparameters for training in our experiments.

| Category | Setting |
|---|---|
| Max sequence length | 25K |
| RoPE $\theta$ | 1,000,000 |
| Dataset | open-thoughts/OpenThoughts3-1.2M (Math Subset) |
| Training objective | Completion-only loss (NLL) |
| Optimizer | AdamW |
| Learning rate | $1 \times 10^{-5}$ |
| LR scheduler | Cosine with minimum LR |
| Min LR rate | 0.01 |
| Warmup ratio | 0.03 |
| Weight decay | 0.00 |
| Per-device batch size | 1 |
| Grad. accumulation | 2 steps |
| Effective batch size | 2 (per device) |
| Gradient checkpointing | Enabled |
| Precision | BF16 enabled (dtype=bfloat16) |

the 4-bit configuration, DGX Spark does not support the native NVFP4 KV cache path in TensorRT-LLM; therefore, we use an FP8 KV cache on DGX Spark (W4A4KV8), while B200 uses the native 4-bit KV cache path (W4A4KV4). To stress memory and capture long-generation serving behavior, we always allow batching up to the maximum batch size of 256 and configure the benchmark to use the largest feasible batch size under a KV cache memory–saturated regime, i.e., we push KV cache allocation toward the memory limit so that batch size is primarily determined by KV cache capacity at longer output lengths. Finally, to implement Q-FIT, we disable RoPE fusion in model implementation and apply the required shifting after RoPE; as shown in Fig. 6, this introduces a negligible runtime overhead compared to native NVFP4, consistent with the small throughput reduction reported in the main text.

### D.4. Ablation Studies

*Table 12.* AIME-90 (2022-2024) accuracy comparison between SEM binary mask and SEM soft weighting under MXFP4 W4A4 TAQ with Q-FIT. **Bold** denotes the better result at each fine-tuning token budget (Model: R1-Qwen-14B).

| Total Fine-Tuning Tokens | Binary Mask | Soft Weighting |
|---|---|---|
| 140M | 60.14 | **61.81** |
| 210M | 63.19 | **65.14** |

**SEM: binary mask vs. soft weighting.** Table 12 shows that soft weighting consistently outperforms a hard binary mask, yielding higher AIME accuracy. Even within low-entropy tokens, aggressively enforcing entropy minimization can be harmful for tokens near the entropy boundary. A hard binary mask may suppress necessary flexibility for such tokens, whereas soft weighting provides smoother control over the regularization strength.

**Multi-epoch QAT vs. SEM.** Table 13 compares simply increasing aligned QAT tokens via multi-epoch TAQ with adding ReQAT. Even with up to 5 epochs (350M tokens), multi-epoch TAQ yields only marginal or even degraded performance, whereas ReQAT achieves higher accuracy even with a single epoch. This indicates that anchor robustness is not obtained by "more QAT" alone; the learning signal must explicitly emphasize low-entropy anchor decisions.

**ReQAT vs. non-reasoning benchmarks.** Table 14 reports results on the MMLU benchmark. Compared to AIME-120, MMLU exhibits substantially smaller accuracy degradation under PTQ, indicating that non-reasoning tasks are less sensitive to quantization. Consequently, both FT and ReQAT provide only marginal improvements over PTQ on MMLU, despite delivering clear gains on reasoning-intensive benchmarks. These results suggest that ReQAT provides its most pronounced benefits on reasoning-intensive benchmarks, while maintaining comparable performance to PTQ and BF16 baselines on non-reasoning tasks such as MMLU.

**Decoding strategy vs. Quantization.** Table 15 shows a related decoding effect. Greedy decoding, which always selects the top-1 token, reduces the PTQ accuracy drop relative to stochastic sampling, but also lowers absolute accuracy. This suggests

*Table 13.* Multi-epoch TAQ vs. ReQAT on AIME (R1-Llama-8B).

| TAQ Tokens | Multi-epoch TAQ (1-5 epochs) | | | | | ReQAT |
|---|---|---|---|---|---|---|
| | 70M | 140M | 210M | 280M | 350M | 70M |
| AIME Acc. | 38.34 | 39.27 | 38.86 | 39.48 | 37.71 | **41.85** |

*Table 14.* Results of ReQAT on MMLU benchmark (model: R1-Qwen-14B).

| Bit-Precision | Method | AIME-120 | MMLU |
|---|---|---|---|
| BF16 | Baseline | 56.83 | 74.96 |
| | FT | 64.79 | 74.84 |
| W4A16 MXFP4 | PTQ | 50.37 | 73.60 |
| | ReQAT | 68.02 | 74.14 |
| W4A4KV4 NVFP4 | PTQ | 50.13 | 72.12 |
| | ReQAT | 65.63 | 73.14 |

*Table 15.* Accuracy (AIME-120) under NVFP4 W4A4KV4 on R1-Llama-8B, measured relative to the corresponding full-precision FT baseline. $T$ denotes temperature.

| Decoding Setting | Decoding Params | Method | FT Epochs ($k$) | | | | |
|---|---|---|---|---|---|---|---|
| | | | 1 | 2 | 3 | 4 | 5 |
| **Sampling** | $T$=0.6, top-p=0.95 | FT | 37.61 | 48.54 | 51.46 | 51.36 | 48.75 |
| | | FT+PTQ | 26.56 | 33.75 | 36.25 | 35.94 | 34.06 |
| | | $\Delta$ (FT − FT+PTQ) | 11.05 | 14.79 | 15.21 | 15.42 | 14.69 |
| **Greedy** | $T$=0, top-k=1 | FT | 23.33 | 35.83 | 44.17 | 45.84 | 43.33 |
| | | FT+PTQ | 15.84 | 24.17 | 33.33 | 30.84 | 35.00 |
| | | $\Delta$ (FT − FT+PTQ) | 7.50 | 11.66 | 10.84 | 15.00 | 8.33 |

that while eliminating sampling errors mitigates some quantization-induced failures, decoding strategy alone is insufficient to fully recover reasoning performance.

*Table 16.* Ablation results of $\lambda$ in SEM (R1-Llama-8B).

| Q-FIT | $\lambda_{\text{SEM}}$ | GSM-8K | MATH-500 | AIME-120 | Avg. |
|---|---|---|---|---|---|
| – | – | 89.38 | 89.80 | 38.34 | 72.51 |
| ✓ | – | 89.86 | 90.72 | 40.32 | 73.63 |
| ✓ | 0.03 | 89.84 | 89.68 | 41.46 | 73.66 |
| ✓ | 0.1 | 89.95 | 90.53 | 41.85 | **74.11** |
| ✓ | 0.5 | 89.69 | 90.45 | 41.88 | 74.01 |

**Effect of SEM strength.** Table 16 ablates the SEM regularization strength $\lambda$ with Q-FIT enabled. Across a broad range of values, SEM yields consistent improvements over the no-SEM baseline, indicating that its benefit is not highly sensitive to precise hyperparameter tuning. A moderate setting ($\lambda_{\text{SEM}} = 0.1$) achieves the best average accuracy across GSM8K, MATH-500, and AIME-120, while both smaller and larger values perform comparably with only minor variation.

*Table 17.* Q-FIT vs. RHT on top of TAQ (R1-Qwen-14B).

| Bit-Precision | Transformation Method | Total Fine-tuning Tokens | | | |
|---|---|---|---|---|---|
| | | 140M | 210M | 280M | 350M |
| NVFP4 W4A4KV4 | – | **60.32** | 60.42 | 63.13 | 63.12 |
| | Block Hadamard | 58.34 | 62.50 | 63.02 | 62.71 |
| | Q-FIT | 59.79 | **63.44** | **65.94** | **65.21** |

**Q-FIT vs. rotation in NVFP4.** Table 17 compares Q-FIT against randomized Hadamard transformation (RHT) (Ashkboos et al., 2024) under NVFP4 W4A4KV4. Q-FIT yields consistently higher accuracy than RHT.

