# OpenReview forum: "ReQAT: Achieving Full-Precision Reasoning Accuracy with 4-bit Floating-Point Quantization-Aware Training"
_ICML.cc/2026/Conference — ICML 2026 spotlight_

### Official Review · Reviewer_6gGy · 2026-03-10

**Soundness:** 3
**Presentation:** 3
**Significance:** 3
**Originality:** 3
**Overall Recommendation:** 5
**Confidence:** 3

**Summary:**

Overall, the submission's principal objective pertains to enabling efficient FP4 deployment of Large Reasoning Models (LRMs) without sacrificing reasoning accuracy. Overall, the authors explore a central theme of entropy-aware quantization-aware training, proposing ReQAT — comprising TAQ, SEM, and Q-FIT — to overcome severe reasoning degradation under W4A4KV4 quantization. While the entropy-centric analytical framework is novel and the method design is well-motivated, the paper contains a critical evaluation methodology concern that undermines confidence in the reported results.

**Compliance With Llm Reviewing Policy:**

Affirmed.

**Final Justification:**

The paper is technically sound, original and significant. My final justification is to accept this paper. My main concern is that about the suspiciously low baseline accuracy and ablation study and the rebuttal solved my concern.

**Key Questions For Authors:**

Please check weakness

**Limitations:**

yes

**Strengths And Weaknesses:**

Strengths
S1. Novel and well-supported core insight. The identification of low-entropy tokens as the dominant failure mode under FP4 quantization is a genuinely useful contribution. The evidence chain from mixed-precision routing (Fig. 2(c-d)) through logit-noise injection (Fig. 2(e)) to tail-mass ratio analysis (Fig. 2(f)) is logically coherent and methodologically creative.
S2. Well-integrated method with clear ablations. TAQ, SEM, and Q-FIT each address distinct failure modes, and Tables 1, 4, and 5 credibly quantify each component's contribution. Table 4's trace-alignment ablation (~5% accuracy gap) is particularly convincing.
S3. Credible hardware throughput evaluation. End-to-end throughput benchmarks on DGX Spark and B200 with memory breakdown analysis (Fig. 6) provide genuine system-level validation.
Weaknesses
W1. Suspiciously low baseline accuracy
The paper reports a BF16 baseline of 56.83% AIME accuracy for R1-Qwen-14B. However, DeepSeek's official evaluation of this same model reports approximately 69.7% on AIME 2024 — a gap of roughly 13 percentage points. Please provide some explanation and especially the detailed evaluation setting you use.
W2. Missing accuracy vs. sequence length analysis.
A core challenge in quantizing reasoning models is error accumulation over long generation sequences: quantization noise at early token positions corrupts intermediate results, and these errors compound through subsequent reasoning steps. Accuracy as a function of generation length is therefore a critical evaluation that directly probes the practical viability of the proposed method.

---

> ### Author Rebuttal · Authors · 2026-03-30
>
> Thank you for the careful reading of our paper and for the constructive feedback.
>
> ---
>
> > **W1.** Suspiciously low baseline accuracy. The paper reports a BF16 baseline of 56.83% AIME accuracy for R1-Qwen-14B. However, DeepSeek's official evaluation of this same model reports approximately 69.7% on AIME 2024 — a gap of roughly 13 percentage points.
>
> We appreciate the reviewer's careful scrutiny of our baselines. The perceived accuracy gap is entirely due to a difference in the evaluation sets: **we report the average across AIME-120** (2022–2025, 120 samples), whereas **DeepSeek’s official 69.70% figure is based exclusively on AIME-2024** (30 samples).
>
> When evaluated under the **exact same AIME-2024 setting**, our BF16 baseline achieves 68.33%, which **closely aligns with DeepSeek's official report**. To clarify this, we provide a year-by-year breakdown of the AIME accuracy:
>
> |Method|2022|2023|2024|2025|Avg (AIME-120)|
> |-|-|-|-|-|-|
> |BF16 Baseline (our reproduce)|56.67|52.92|68.33|49.38|56.83|
> |BF16 Baseline (DeepSeek official)|-| -|69.70|-|-|
> |BF16 Baseline (reported in [R1])|-|-|-|-|57.50|
> |BF16 Baseline (reported in [R2])|-|-|-|-|54.70|
> |BF16 FT|66.25|64.58|72.08|58.96|65.47|
> |PTQ|48.33|49.58|59.58|42.92|50.10|
> |FT+PTQ|55.00|56.67|58.33|52.92|55.73|
> |QAT|60.83|56.25|67.50|50.83|58.85|
> |**ReQAT**|67.92|65.42|71.25|59.17|**65.94**|
>
> **Table R1.** Year-wise AIME accuracy (NVFP4 W4A4KV4 R1-Qwen-14B). For fine-tuned methods, we report the best accuracy across budgets of 140M–350M tokens.
>
> Key Takeaways:
> - **No Baseline Discrepancy**: Our 68.33% on AIME-2024 demonstrates that **our baseline setup is fundamentally sound and matches official capabilities**.
> - **AIME-120 is More Rigorous**: We chose AIME-120 because it offers a **larger sample size (120 vs. 30)** and includes notably **harder distributions (e.g., AIME-2025)**. Our average baseline of 56.83% is **highly consistent with recent literature evaluating this exact benchmark** (57.50% in [R1] and 54.70% in [R2]).
> - **ReQAT's Robustness**: Crucially, across all years, **ReQAT consistently closes the quantization gap** and even slightly exceeds the BF16 Full-FT accuracy.
>
> **Evaluation details.** We use the lighteval framework for AIME. Decoding follows DeepSeek recommendations: temperature = 0.6, top-p = 0.95, maximum sequence length = 32K.
>
> ---
>
> > **W2.** Missing accuracy vs. sequence length analysis. A core challenge in quantizing reasoning models is error accumulation over long generation sequences: quantization noise at early token positions corrupts intermediate results, and these errors compound through subsequent reasoning steps. Accuracy as a function of generation length is therefore a critical evaluation that directly probes the practical viability of the proposed method.
>
> We sincerely thank the reviewer for highlighting the critical challenge of error accumulation over extended reasoning sequences. To directly address this, we conducted an in-depth analysis of 960 AIME responses, categorizing them by generation length. This allows us to explicitly track how well each method resists the compounding effects of quantization noise as reasoning chains grow longer.
>
> |Length Range|BF16 Baseline|BF16 FT|PTQ|FT+PTQ|QAT|ReQAT|
> |-|-|-|-|-|-|-|
> |0–8K|92.1/355|**99.4**/166|84.6/363|98.8/170|**100.0**/155|98.9/159|
> |8–16K|55.1/303|**91.5**/318|48.5/297|80.5/302|88.5/288|**91.7**/333|
> |16–24K|21.9/178|**57.4**/230|14.7/204|42.3/213|50.9/222|**55.7**/219|
> |24–32K|12.9/124|**17.5**/246|6.3/96|12.7/275|14.2/295|**19.3**/249|
>
> **Table R2**. Accuracy and sample count (in parentheses) grouped by each method's generated response length (NVFP4 W4A4KV4 R1-Qwen-14B).
>
> Key Takeaways:
> - **The Compounding Error of PTQ**: As expected, both PTQ and FT+PTQ suffer severe degradation as response length increases, highlighting its vulnerability to error cascading in complex problems.
> - **Reasoning Depth vs. Stability**: Fine-tuning (BF16 FT) naturally encourages deeper, more extensive reasoning, producing roughly 2× more samples in the 24K–32K range than the baseline. However, standard QAT struggles to maintain stability at these extended lengths, showing a noticeable drop compared to the BF16 FT model.
> - **ReQAT’s Robustness at Scale**: ReQAT successfully breaks the chain of error accumulation. It not only sustains a **high volume of long responses** (preserving reasoning depth) but also achieves the **highest accuracy in the most demanding 24K–32K** tier (19.3%). This clearly demonstrates ReQAT's superior robustness to cumulative quantization noise without trading off reasoning capability.
>
> We will incorporate this detailed sequence-length analysis into the revised manuscript and release our model weights to ensure full reproducibility.
>
> [R1] A Case Study of Selected PTQ Baselines for Reasoning LLMs on Ascend NPU, 2026
> [R2] Quantization Hurts Reasoning? An Empirical Study on Quantized Reasoning Models, COLM'25

---

> > ### Author Rebuttal · Reviewer_6gGy · 2026-04-01
> >
> > The experiment results fully solve my concerns.

---

> > > ### Author Response · Authors · 2026-04-08
> > >
> > > We thank the reviewer for the careful reading and constructive feedback, especially for recognizing the novelty of our entropy-centric analysis, the design of ReQAT, and the system-level throughput evaluation.
> > >
> > > We are glad that our clarifications and additional analyses helped resolve the reviewer’s concerns and strengthen the rigor of our evaluation. In particular, we clarified that the apparent baseline gap comes from the difference between the AIME-120 and AIME-2024 evaluation settings, and the year-wise breakdown showed that our reproduced BF16 baseline closely matches the official result under the same setting. We also added a length-wise analysis showing that ReQAT better maintains accuracy as reasoning traces grow longer, which supports its robustness to cumulative quantization error.
> > >
> > > We will incorporate these clarifications and analyses in the final version. We also plan to release our ReQAT framework and trained model weights to support reproducibility and contribute to further progress in this area. We thank the reviewer again for the constructive feedback and for the time and care taken in evaluating our work.

---

### Official Review · Reviewer_6N1c · 2026-03-12

**Soundness:** 4
**Presentation:** 4
**Significance:** 4
**Originality:** 4
**Overall Recommendation:** 5
**Confidence:** 4

**Summary:**

This paper identifies a crucial problem: the unique sensitivity of LRMs to quantization, and provides a thorough empirical analysis pinpointing the root cause, which is low-entropy token corruption, and proposes a novel, well-engineered solution (ReQAT) that is both theoretically motivated and practically effective. The empirical results are outstanding, demonstrating that a quantized model can surpass its full-precision counterpart in accuracy while achieving massive (3-4x) throughput gains on real hardware.

**Compliance With Llm Reviewing Policy:**

Affirmed.

**Final Justification:**

Given the author's responses, I recommend accept this paper.

**Key Questions For Authors:**

1. The SEM loss selectively minimizes entropy at low-entropy positions. Is there a risk of over-confident predictions or a decrease in the model's ability to explore alternative paths?
2. Do you have any preliminary evidence or hypotheses on whether this finding generalizes to other reasoning-heavy domains like code generation?

**Limitations:**

Yes

**Strengths And Weaknesses:**

Strengths:
1. The paper's core contribution is the identification that low-entropy tokens (digits, operators) are the primary point of failure when quantizing LRMs to 4-bit. This insight is both surprising and profound.
2. This paper also builds a complete and effective framework, ReQAT, to solve it, developing three components: TAQ, SEM and Q-FIT.
3. The results in Table 1 are remarkable. ReQAT not only recovers the accuracy lost to quantization but surpasses the BF16 baseline and even BF16 full fine-tuning on the challenging AIME benchmark under the extreme NVFP4 W4A4KV4 setting.

Weaknesses:
1. The success of TAQ relies on having a high-quality dataset of reasoning traces to align the two training stages. For a new reasoning domain without such a dataset, the benefits of TAQ might be harder to realize.
2. The core insight is demonstrated for mathematical reasoning, as shown in evaluation benchmarks. It would be valuable to know if this insight holds for other forms of reasoning, such as code generation.

---

> ### Author Rebuttal · Authors · 2026-03-30
>
> We appreciate the reviewer’s careful understanding of our work.
>
> ---
>
> > **W1.** TAQ reliance on high-quality reasoning traces.
>
> While we agree that high-quality data universally benefits fine-tuning, TAQ's effectiveness is fundamentally mechanism-driven rather than dataset-dependent. The core benefit of TAQ arises from revisiting the same traces, which forces the optimization process to reallocate its learning signals toward vulnerable low-entropy tokens, rather than learning new high-entropy reasoning paths.
>
> To provide technical evidence, we analyzed the token-level learning dynamics via embedding gradients. Let $E \in \mathbb{R}^{T \times D}$ denote the token embedding matrix and $G = \nabla_E \mathcal{L}$ the gradient of the loss with respect to $E$. We define the per-token gradient magnitude $s\_t$ and the gradient contribution ratio of low-entropy tokens $C\_{\text{low}}$ as:
> $$
> s\_t = \|G\_{t,:}\|\_2^2, \quad C\_{\text{low}} = \frac{\sum\_{t \in \mathcal{I}\_{\text{low}}} s\_t}{\sum\_{t \in \mathcal{I}} s\_t}
> $$
> where $\mathcal{I}$ denotes all tokens and $\mathcal{I}_{\text{low}}$ the low-entropy subset.
>
> https://imgur.com/esfoIHU
> **Figure R1.** Gradient contribution ratio of low-entropy tokens.
>
> As demonstrated in Fig. R1:
> - **Focusing the Signal (TAQ)**: Revisiting aligned traces during QAT consistently increases $C_{\text{low}}$. Because the model has already learned the high-entropy reasoning structure during the Stage-1 BF16 FT, the QAT gradients naturally shift their focus to correcting low-entropy tokens.
> - **Diluting the Signal (Misaligned Data)**: When the QAT data is misaligned, this effect diminishes, indicating weaker learning signal on low-entropy tokens critical for quantized reasoning.
>
> In short, TAQ does not strictly require a "magical" or massive reasoning dataset; it merely requires consistent supervision across stages to activate this gradient reallocation. Therefore, this mechanism is highly generalizable and will extend to any new structured domain as long as the initial FT dataset can be reused for the QAT stage.
>
> ---
>
> > **W2&Q2.** Generalization beyond math reasoning.
>
> We thank the reviewer for this valuable suggestion. We validate this generalization to code generation through both downstream accuracy on LiveCodeBench and token-level entropy analysis.
>
> |Method|MXFP4 W4A16|NVFP4 W4A4KV4|
> |-|-|-|
> |BF16 Baseline|51.68|51.68|
> |BF16 FT|53.68|53.68|
> |PTQ|48.88|47.99|
> |QAT|52.01|50.68|
> |**ReQAT$_\text{TQ}$**|53.82|53.08|
> |**ReQAT$_\text{TQS}$**|**54.52**|**53.59**|
>
> **Table R1**. Accuracy on LiveCodeBench (R1-Qwen-14B).
>
> As shown in Table R1, even without code-specific QAT traces, ReQAT outperforms standard PTQ and QAT baselines. Crucially, incorporating SEM (ReQAT$_\text{TQS}$) further improves accuracy, enabling the 4-bit model to match or slightly exceed the BF16 FT model.
>
> https://imgur.com/tBidzgf
> **Figure R2.** Visualization of low- and high-entropy tokens extracted from LLM responses for math and coding tasks.
>
> To understand why this zero-shot transfer is effective, we analyzed 1.5M tokens from model-generated code outputs. As shown in Fig. R2, low-entropy tokens correspond to similar types of structured elements (e.g., digits, operators) across domains. Because the structural vulnerability to quantization noise is shared across domains, ReQAT's stabilizing benefits directly transfer.
>
> ---
>
> > **Q1.** Risk of over-confident predictions with SEM.
>
> We observe that entropy minimization (EM), when applied without careful control, can degrade reasoning accuracy, reflecting risks such as over-confident predictions and reduced exploration. SEM mitigates this risk through two mechanisms: selective application to low-entropy tokens and soft weighting to avoid overly rigid updates.
>
> |Method|Acc|
> |-|-|
> |Baseline (TAQ + Q-FIT, No SEM)|61.36|
> |EM on low-entropy tokens (SEM)|**65.00**|
> |EM on high-entropy tokens|59.90|
>
> **Table R2.** AIME-120 results on W4A16 MXFP4 R1-Qwen-14B.
>
> To verify this, we apply EM to different token groups in Table R2. Applying EM to high-entropy tokens degrades accuracy, indicating suppressed exploration.
>
> Even when applied to low-entropy tokens, EM can still introduce similar risks if enforced too rigidly, particularly for tokens near the entropy boundary (i.e., relatively higher-entropy tokens within the low-entropy group). A binary mask can mis-handle such tokens and suppress necessary flexibility; soft weighting mitigates this issue.
>
> |FT Tokens|No SEM|SEM w/ Binary Mask|SEM w/ Soft Weighting|
> |-|-|-|-|
> |140M|61.53|60.14|**61.81**|
> |210M|64.44|63.19|**65.14**|
>
> **Table R3.** AIME-90 (2022–2024) accuracy under MXFP4 W4A4.
> - As shown in Table R3, binary masking underperforms even without SEM, while soft weighting consistently improves performance.
> - Overall, SEM mitigates this risk through **selective application** and **soft weighting**, preserving exploration.
>
> We will incorporate these results and analyses into the revised version.

---

> > ### Author Rebuttal · Reviewer_6N1c · 2026-04-03
> >
> > My concerns have been adequately addressed.

---

> > > ### Author Response · Authors · 2026-04-08
> > >
> > > We sincerely thank the reviewer for the thorough and insightful evaluation, especially for highlighting our identification of low-entropy tokens as a dominant failure mode under FP4 quantization and the effectiveness of the ReQAT framework.
> > >
> > > We are glad that our additional analyses helped address the reviewer’s concerns and further strengthen the paper. In particular, the gradient contribution analysis showed that revisiting aligned traces during QAT reallocates learning signal toward vulnerable low-entropy tokens, supporting that TAQ benefits come from its design rather than from a dataset-specific effect. The LiveCodeBench results supported generalization beyond mathematical reasoning, and the SEM ablations clarified the benefit of selective and softly weighted entropy minimization.
> > >
> > > We also plan to release our ReQAT framework and trained model weights to support reproducibility and further research on efficient reasoning model deployment. We thank the reviewer again for the thoughtful feedback and for the time and effort invested in evaluating our work.

---

### Official Review · Reviewer_tUnr · 2026-03-13

**Soundness:** 3
**Presentation:** 4
**Significance:** 3
**Originality:** 3
**Overall Recommendation:** 5
**Confidence:** 2

**Summary:**

This paper studies the degradation of reasoning performance in large reasoning models when deployed under aggressive FP4 quantization. The authors observe that quantization errors disproportionately affect low-entropy tokens, such as numbers and symbols, which often correspond to critical computation steps in chain-of-thought reasoning. Errors on these tokens can easily propagate and lead to incorrect final answers.

Based on this observation, the paper proposes ReQAT (Reasoning-centric Quantization-Aware Training), a framework designed to improve reasoning robustness under FP4 quantization. The framework consists of three main components: Trace-Aligned QAT (TAQ), which reuses reasoning traces during quantization-aware training; Selective Entropy Minimization (SEM), which encourages confident predictions on low-entropy tokens; and Q-FIT, a quantization-friendly initialization method for KV caches.

Experiments on several mathematical reasoning benchmarks, including AIME, MATH, and GSM8K, show that ReQAT can significantly improve reasoning accuracy under FP4 quantization and in some cases even outperform BF16 fine-tuned models while achieving higher inference efficiency.

**Compliance With Llm Reviewing Policy:**

Affirmed.

**Key Questions For Authors:**

No

**Limitations:**

Yes, as Impact Statement

**Strengths And Weaknesses:**

Strengths and Weaknesses

Strengths
	1.	The problem addressed is practically important.
As large reasoning models become increasingly expensive to deploy, reducing inference cost through low-bit quantization is an important research direction. Studying how quantization affects reasoning performance is therefore both timely and relevant.
	2.	The paper provides an interesting empirical observation.
The analysis showing that quantization errors disproportionately impact low-entropy tokens (e.g., numbers and symbols) offers a useful perspective on why reasoning models degrade under low-precision settings.
	3.	The proposed framework is well motivated and reasonably designed.
The components of ReQAT (TAQ, SEM, and Q-FIT) target different aspects of the quantization problem, including reasoning trace alignment, confidence calibration on key tokens, and KV cache quantization stability.
	4.	The empirical evaluation is reasonably thorough.
The paper reports results across multiple reasoning benchmarks and provides ablation studies that help illustrate the contribution of different components.

⸻

Weaknesses

Overall, the paper is well executed and I did not notice major technical issues. The following comments are mainly suggestions that could further strengthen the work.
	1.	Experimental coverage could be expanded.
The current experiments mainly focus on mathematical reasoning benchmarks (e.g., AIME, MATH, GSM8K). Evaluating the method on additional types of reasoning tasks or different model scales could further demonstrate the generality of the approach.

---

> ### Author Rebuttal · Authors · 2026-03-30
>
> Thank you for the positive assessment and constructive suggestion.
>
> ---
>
> > **W1.** Experimental coverage could be expanded. The current experiments mainly focus on mathematical reasoning benchmarks (e.g., AIME, MATH, GSM8K). Evaluating the method on additional types of reasoning tasks or different model scales could further demonstrate the generality of the approach.
>
> We thank the reviewer for the helpful suggestion. Following the reviewer’s suggestion, we additionally evaluate models trained on mathematical reasoning traces on an out-of-domain coding benchmark, LiveCodeBench. We validate this generalization to code generation through both downstream accuracy on LiveCodeBench and token-level entropy analysis across domains.
>
>
> | Method | MXFP4 W4A16 | NVFP4 W4A4KV4 |
> |---|---|---|
> | BF16 Baseline | 51.68 | 51.68 |
> | BF16 Full-FT | 53.68 | 53.68 |
> | PTQ | 48.88 | 47.99 |
> | QAT | 52.01 | 50.68 |
> | **ReQAT$_\text{TQ}$** | 53.82 | 53.08 |
> | **ReQAT$_\text{TQS}$** | **54.52** | **53.59** |
>
> **Table R1.** Accuracy on LiveCodeBench (R1-Qwen-14B).
>
> - **ReQAT consistently improves over QAT** and achieves accuracy **comparable to BF16 full fine-tuning under FP4** quantization.
> - Incorporating SEM (ReQAT$\_\text{TQS}$) further improves accuracy over ReQAT$\_\text{TQ}$, indicating that **encouraging more confident predictions on low-entropy tokens is also effective in this domain**.
>
> https://imgur.com/tBidzgf
> **Figure R1.** Visualization of low- and high-entropy tokens in mathematical (AIME) and coding (LiveCodeBench) outputs.
>
> To understand why this zero-shot transfer is effective, we analyzed 1.5M tokens from model-generated code outputs (Fig. R1). We find that low-entropy tokens correspond to similar structured elements across both domains, such as digits, operators, and symbols. This structural similarity suggests that the token-level mechanisms used in ReQAT can generalize beyond mathematical reasoning.
>
> These results support the generalization of ReQAT beyond mathematical reasoning and suggest its applicability to other structured domains.
>
> We thank the reviewer again for the helpful suggestion. We will include these additional results in the revised version.

---

> > ### Author Rebuttal · Reviewer_tUnr · 2026-04-02
> >
> > I am not familiar with the field.
> >
> > But I think your work is solid, and the results are promising.

---

> > > ### Author Response · Authors · 2026-04-08
> > >
> > > We thank the reviewer for the positive assessment, especially for recognizing the practical importance of efficient FP4 deployment for large reasoning models and our analysis of the vulnerability of low-entropy tokens under quantization.
> > >
> > > We are glad that the additional LiveCodeBench results and cross-domain token-level entropy analysis helped address the concern on generalization. These results support that the same types of structurally critical low-entropy tokens remain vulnerable across domains beyond mathematical reasoning.
> > >
> > > We will incorporate these results in the final version. We also plan to release our ReQAT framework and trained model weights to support reproducibility and further research on efficient reasoning model deployment. We thank the reviewer again for the thoughtful feedback and for the time and care invested in reviewing our work.

---

### Decision · Program_Chairs · 2026-04-30

**Decision:**

Accept (spotlight)

**Comment:**

This paper aims at an agressive FP4 quantization of reasoning LMs. The core insight is that low-entropy tokens (digits and operators) are the primary failure mode under aggressive quantization. Based on this insight, this paper proposes trace-aligned training, selective entropy minimization, and quantization-aware initialization. Experiment results are very strong, sometimes even outperforming BF16. Limitations are mainly that experiments are mostly focused on math (later authors added code). Given the practical implications of this level of quantization for the deployment of LMs, and that reviewers are unanimous (after rebuttal) to accept this paper, I recommend to accept this paper.